



# First assessment of Aeolus L2A particle backscatter coefficient retrievals in the Eastern Mediterranean

Antonis Gkikas[1], Anna Gialitaki[1,5,6], Ioannis Binietoglou[1], Eleni Marinou[1], Maria Tsichla[1], Nikolaos Siomos[1], Peristera Paschou[1,5], Anna Kampouri[1,7], Kalliopi Artemis Voudouri[1,5], Emmanouil Proestakis[1], Maria Mylonaki[2], Christina-Anna Papanikolaou[2], Konstantinos Michailidis[5], Holger Baars[3], Anne Grete Straume[4], Dimitris Balis[5], Alexandros Papayannis[2], Tomasso Parrinello[8] and Vassilis Amiridis[1]

[1]Institute for Astronomy, Astrophysics, Space Applications and Remote Sensing, National Observatory of Athens, Athens, Greece

[2]Laser Remote Sensing Unit, Department of Physics, National and Technical University of Athens, Athens, Greece

[3]Leibniz-Institut für Troposphärenforschung e.V., Leipzig, Germany

[4]European Space Agency (ESA/ESTEC), Noordwijk, Netherlands

[5]Laboratory of Atmospheric Physics, Aristotle University of Thessaloniki, Thessaloniki, Greece

[6]Department of Physics and Astronomy, University of Leicester, Leicester, United Kingdom

[7]Department of Meteorology and Climatology, School of Geology, Aristotle University of Thessaloniki, 54124 Thessaloniki, Greece

[8]European Space Agency (ESA/ESRIN), Frascati, Italy

Corresponding author: Antonis Gkikas (agkikas@noa.gr)

## Abstract

Since 2018, the Aeolus satellite of the European Space Agency (ESA) acquires wind HLOS (horizontal line-of-sight) profiles throughout the troposphere and up to the lower stratosphere, filling a critical gap of the Global Observing System (GOS). Aeolus, carrying ALADIN, the first UV HSRL Doppler lidar ever placed in space, along with wind HLOS profiles provides also vertically resolved optical properties of particulates (aerosols, hydrometeors). The present study focuses on the assessment of Aeolus L2A particulate backscatter coefficient, retrieved by the Standard Correct Algorithm (SCA), in the Eastern Mediterranean, a region hosting a variety of aerosol species. Ground-based retrievals acquired by lidar instruments operating in Athens (capital of Greece), Thessaloniki (north Greece) and Antikythera (southwest Greece) serve as reference. All lidar stations provide routine measurements to the PANACEA (PANhellenic infrastructure for Atmospheric Composition and climatE chAnge) network. A set of ancillary data including sunphotometric observations (AERONET), reanalysis products (CAMS, MERRA-2), satellite observations (MSG-SEVIRI, MODIS-Aqua) and backward trajectories (FLEXPART) are utilized towards an optimum



characterization of the probed atmospheric conditions under the absence of a classification scheme in
Aeolus profiles. First, emphasis is given on the assessment of Aeolus L2A backscatter coefficient
under different aerosol scenarios over Antikythera island. Due to the misdetection of the cross-polar
component of the backscattered lidar signal, Aeolus underestimates backscatter by up to 33% when
non-spherical mineral particles are recorded ($10^{th}$ July 2019). A very good performance is revealed
on $3^{rd}$ July 2019, when homogeneous loads of fine spherical particles are confined below 4 km. The
level of agreement between spaceborne and ground-based retrievals varies with altitude when aerosol
layers, composed of particles of different origin, are stratified ($8^{th}$ July 2020, $5^{th}$ August 2020).
According to the statistical assessment analysis for 46 identified cases, it is revealed a poor-to-
moderate performance for the unfiltered (aerosols plus clouds) Aeolus profiles which improves
substantially when cloud contaminated profiles are excluded from the collocated sample. This
positive tendency is evident at both Aeolus vertical scales (regular, 24 bins and mid-bin, 23 bins) and
it is justified by the drastic reduction of the bias and root-mean-square-error scores. In vertical, Aeolus
performance downgrades at the lowermost bins (attributed to either the surface reflectance or the
increased noise levels for the Aeolus retrievals and to the overlap issues for the ground-based
profiles). Among the three PANACEA stations, the best agreement is found at the remote site of
Antikythera with respect to the urban sites of Athens and Thessaloniki. Finally, all key Cal/Val
aspects necessary for future relevant studies, the recommendations for a possible Aeolus follow-on
mission and an overview of the ongoing related activities are thoroughly discussed.

## 1. Introduction

Atmospheric aerosols constitute a critical component of the Earth system by acting as a major

climatic driver (Charlson et al., 1992; Boucher et al., 2013; Li et al., 2022) whereas upon deposition
they can affect terrestrial (Okin et al., 2004) and marine ecosystems (Jickells et al., 2005; Li et al.,
2018). It is also well documented that they affect several anthropogenic activities with concomitant
economic impacts (Middleton et al., 2018; Kosmopoulos et al., 2018). In addition, the accumulation
of aerosols at large concentrations causes a degradation of air quality (Kanakidou et al., 2011) with
adverse health effects (Pöschl, 2005; Lelieveld et al., 2015) that can increase the mortality rates
(Health Effects Institute, 2019; Pye et al., 2021). Therefore, their multifaceted role in
multidisciplinary research fields highlights the growing scientific concern in understanding and
describing the emission, removal, and transport mechanisms governing airborne particles' life cycle.
Due to their pronounced heterogeneity, aerosol burden exhibits a remarkable spatiotemporal
variability thus imposing deficiencies in adequately depicting its features and constraints towards a
robust assessment of the induced impacts.



Passive satellite sensors, providing columnar retrievals of aerosol optical depth (AOD), have
been able to reproduce adequately aerosol loads' features across various spatiotemporal scales as it
has been justified via the assessment of AOD versus corresponding sun-photometric measurements
(e.g., Wei et al., 2019). Nevertheless, the main drawback arises from the sensors' inability to provide
information on the vertical distribution of aerosols within the atmosphere. This deficiency hampers a
reliable quantification of suspended particles within the planetary boundary layer (PBL), related to
health impacts, the three-dimensional reproduction of transported loads in the free troposphere, linked
to aerosol-cloud-radiation interactions and associated impacts on atmospheric dynamics (Perez et al.,
2006; Gkikas et al., 2018; Haywood et al., 2021), as well as to monitor stratospheric long-lived
plumes affecting aerosol-chemistry interactions and perturbing the radiation fields (Solomon et al.,
2022). On the contrary, ground-based lidars, relying on active remote sensing techniques, obtain
vertical profiles of aerosol optical properties at high vertical and temporal resolution, through multi-
wavelength and polarization measurements, as well as the geometric features of the particles' layers.
Such observations are performed either at networks distributed across Europe (EARLINET;
Papalardo et al., 2014; PollyNET; Baars et al., 2016; Engelmann et al., 2016), United States
(MPLNET; Campbell et al., 2002), Asia (AD-NET; Sugimoto et al., 2014) and South America
(LALINET; Guerrero-Rascado et al., 2016), or at dedicated experimental campaigns (Ansmann et al.,
2011; Weinzierl et al., 2016) or even at open seas (Bohlmann et al., 2018). The reproduction of
aerosols' vertical structure at global (Liu et al., 2008) and regional (Marinou et al., 2017; Proestakis
et al., 2018) scales has been realized through the utilization of measurements acquired by the Cloud-
Aerosol Lidar and Infrared Pathfinder Satellite Observation (CALIOP; Winker et al., 2009) and the
Cloud-Aerosol Transport System (CATS; McGill et al., 2015) mounted on the CALIPSO (Cloud-
Aerosol Lidar and Infrared Pathfinder Satellite Observation) satellite and the International Space
Station (ISS), respectively.
On 22$^{nd}$ August 2018, the European Space Agency (ESA) launched its Earth Explorer wind
mission, Aeolus. It is the first space-based Doppler Wind lidar worldwide, and was a major step
forward for Earth Observations (EO) and atmospheric sciences. The key scientific objective of Aeolus
is to improve numerical weather forecasts and to improve our understanding of atmospheric dynamics
and their associated impacts on climate (Stoffelen et al., 2005; Isaksen and Rennie, 2019; Rennie and
Isaksen, 2019). The first Aeolus wind product assessment, by the Aeolus Data Innovation Science
Cluster (DISC) and Cal/Val teams (e.g. Baars, et al., 2020; Lux et al., 2020; Witschas et al., 2020,
ESA 2019), during the instrument commissioning phase in autumn 2018 demonstrated that Aeolus
could provide physically meaningful winds in near-real-time immediately after the instrument switch-
on in September 2018, demonstrating the space-based Doppler wind lidar principle for the first time.



However, the analyses also revealed issues with ALADIN's instrument performance and on-ground
data processing performance which needed to be mitigated through careful instrument
characterization, instrument adjustments, improved data calibration and on-ground data processor
updates. After about 1.5 years of instrument and algorithm improvements, the Aeolus L2B wind
product was of such good quality that the European Centre for Medium Range Forecasts (ECMWF)
could start operational assimilation (January 2020). In May 2020, the L2B wind product could
therefore be publicly released, which lead to three further European weather forecast institutes starting
operational assimilation of Aeolus winds to Deutsche Wetter Dienst (DWD), Météo-France and the
UK MetOffice. All meteorological institutes found that Aeolus winds had significant positive impact
on the short and medium term forecasts, with the largest impact in remote areas less covered by other
direct wind observations including the tropics, southern hemisphere and polar areas (e.g. ECMWF
2020; Rennie et al., 2021).
The Aeolus Aladin instrument is a high spectral resolution Doppler wind lidar (HSRL),
emitting circularly polarized laser light at 355 nm and observing the co-polarized backscatter from
molecules and particles and hydrometeors in two separate channels (Ansmann et al., 2007; Flamant
et al., 2008). The backscattered light from the surface or top of optically thick clouds up to 30 km
altitude is sampled with a vertical resolution of 24 range bins with a thickness from 250 m up to 2
km. The main mission product is profiles of the horizontally projected line-of- sight winds, and spin-
off products are the backscatter and extinction coefficient profiles from particles and hydrometeors.
In contrast to CALIOP and CATS, ALADIN can retrieve these products without requiring an a priori
assumption of the lidar ratio (S), which is characterized by a remarkable variability among aerosol
types due to its dependency on particles' shape, composition and size distribution (Müller et al.,
2007). However, Aeolus only measures the co-polar part of the atmospheric backscatter and at a
single wavelength. Therefore, it is very challenging to discriminate the atmospheric backscatter
attributed to aerosols or hydrometeors.
A series of errors induced by the instrument, by the retrieval algorithm, or by the type of
scatterers probed by Aladin can affect the product quality. It is therefore necessary to perform
continuous and extensive calibration and validation (Cal/Val) studies utilizing independent reference
measurements (e.g. ground-based, aircraft). This task has been performed by the Aeolus Cal/Val
community, responding to the Aeolus Announcement of Opportunity to perform product calibration
and validation. Such critical tasks are prerequisites to the acceptance of the Mission as "fit for
purpose" as it is underlined in the Aeolus Implementation Cal/Val Plan. In contrast to Aeolus wind
retrievals, a very limited number of studies focused on the quality of the L2A optical properties. Abril-



Gago et al. (2022) performed a statistical validation versus ground-based observations from three
Iberian ACTRIS/EARLINET lidar stations. Baars et al. (2021) reported an excellent agreement
between Aeolus and Polly$^{XT}$ particle backscatter profiles and adequate agreement of extinction and
lidar ratio profiles for a case of long range transport of wildfire smoke particles from California to
Leipzig (Germany).

The current study focuses on the comparison of Aeolus L2A particle backscatter coefficient

profiles against ground-based profile observations acquired at three lidar stations (Antikythera,
Athens, Thessaloniki) contributing to the Greek National Research Infrastructure (RI) PANACEA,
an ACTRIS component, that assures a homogenous quality. All stations are located in the Eastern
Mediterranean, a crossroad of air masses (Lelieveld et al., 2002) carrying particles of different nature.
The broader Greek area encompasses a variety of aerosol species consisting of: (i) pollutants from
industrialized European regions (Gerasopoulos et al., 2003; 2009), (ii) dust aerosols from the nearby
deserts (Balis et al., 2004; Papayannis et al., 2005; Gkikas et al., 2016, Marinou et al., 2017), (iii)
anthropogenic aerosols from urban areas and megacities (Kanakidou et al., 2011), (iv) biomass
burning particles originating in the eastern Europe and the Black Sea (Amiridis et al., 2009; 2010;
2012), (v) smoke aerosols subjected to transport at planetary scale (Baars et al., 2019; Gialitaki et al.,
2020), (vi) sea-salt particles produced by bursting bubbles during whitecap formation attributed to
wind-wave interactions (e.g. Varlas et al., 2021), (vii) biogenic particles such as airborne fungi and
pollen grains (e.g. Richardson et al., 2019) and (viii) volcanic ash mixed with sulfate aerosols ejected
at high altitudes from explosive Etna eruptions (Zerefos et al., 2006, Kampouri et al., 2021).

The manuscript is structured as follows. In Section 2, a brief overview of the Aeolus satellite

and the ALADIN instrument is given. The key elements of the Standard Correct Algorithm (SCA)
are summarized in Section 3. The technical aspects of the ground-based lidars and the regime of
aerosol loads in the surrounding area of the PANACEA stations are presented in Section 4. The
collocation criteria between ground-based and spaceborne profiles are described in Section 5. The
assessment of Aeolus L2A product under various aerosol scenarios and for the whole collocated
sample are discussed in Section 6. The Cal/Val aspects, the recommendations for future relevant
studies and the necessary upgrades on ALADIN observational capabilities and Aeolus L2A data
content are highlighted in Section 7. Finally, the main findings and the conclusions are drawn in
Section 8.




## 2. AEOLUS - ALADIN

A brief description of Aeolus' orbital features, ALADIN's observational geometry and its measurement configuration is given in the current section. This short introduction serves as the starting point for the reader to be familiar with Aeolus' nomenclature. Further details and a more comprehensive overview of the Aeolus satellite mission can be found at ESA technical reports (ESA, 1999; 2008; 2016) and at recently published studies (e.g., Lux et al., 2020; Witschas et al., 2022; Lux et al., 2022).

ESA's Aeolus satellite, named by the 'keeper of winds' according to the Greek mythology (Ingmann and Straume, 2016), flies in a polar sun-synchronous orbit circling the Earth at an altitude of 320 km with a repeat cycle of 7 days (Kanitz et al., 2019a; Straume et al., 2019). The orbital plane forms an angle of 97° with the equatorial plane, the ground track velocity is about 7.2 km/sec and a complete circle around the Earth takes about 90 minutes for each orbit (Lux et al., 2020; Witschas et al., 2020; Straume et al., 2020). Aeolus is flying over the terminator between day and night (dawn/dusk orbit), with its solar panels facing towards the sun direction for minimizing the solar background illumination (Kanitz et al., 2019).

ALADIN, the single payload on the Aeolus satellite platform, is an HSRL lidar (Shipley et al., 1983) equipped with a Nd-YAG laser that emits short laser pulses (≈ 40 to 70 mJ, Witschas et al., 2020) of a circular polarized light at ~355 nm with a 50.5 Hz repetition frequency. The photons that are backscatter from molecules and particulates (aerosols, cloud droplets and ice crystals) at atmospheric altitudes lower than 30 km are captured by a Cassegrain telescope of 1.5 m diameter. The collected photons are directed to the Mie optical channel (Fizeau interferometer) for the analysis of the Doppler shift induced by particulates while the molecular return signals (Rayleigh) are analyzed in two sequentially coupled Fabry–Pérot interferometers (Witschas et al., 2020).

ALADIN provides wind and particulate vertically resolved retrievals along the Line-Of-Sight (LOS) by pointing at a slant angle of 35° off-nadir (see Figure 1 in Flament et al., (2021)) which increases to 37.6°, depending on altitude, due to the curvature of the Earth surface. The instrument detector design allows the sampling of the atmospheric backscatter in 24 vertical bins, with a varying resolution from 0.25 (earth surface) to 2 km (upper atmosphere). The laser pulses are integrated on-board the satellite along the satellite flight direction, to yield measurements of ~3km resolution (integration of ~20 laser pulses). During the on-ground data processing, the measurements are accumulated further to yield an "*observation*" (also called a *Basic Repeat Cycle (BRC)),* which corresponds to a distance of ~90 km. The L2A optical properties product which will be analyzed in the next section, derived by the so-called Standard Correct Algorithm (SCA) (Flament et al., 2021),



are provided at the observation scale (on a horizontal resolution of ~90 km) and are available through
the Aeolus Online Dissemination System (https://aeolus-ds.eo.esa.int).

**3. Standard Correct Algorithm (SCA)**
Aeolus L2A particulate products are derived by three retrieval algorithms, namely the
Standard Correct Algorithm (SCA), the Mie Correct Algorithm (MCA) and the Iterative Correct
Algorithm (ICA) and their full description is provided in the Algorithm Theoretical Baseline
Document (ATBD; Flamant et al., 2021). MCA relies only on the Mie channel returns and the
implementation of the Klett method (Klett, 1981) under the assumption of a universal lidar ratio (~14
sr). On the other hand, ICA works under the assumption of different partial filling of the particles
within the range bin. Finally, there is also the *group* product in which signals of high signal-to-noise
(SNR) ratio are accumulated prior to the implementation of the SCA algorithm. Both ICA and group
products are still under development and they are not recommended to be utilized in scientific studies
(Flament et al., 2021).
Among the aforementioned Aeolus L2A retrieval algorithms, the primary, the most reliable
and mature is the SCA. The SCA product is derived from the measured signals on the Mie and
Rayleigh channels, which are dependent on the instrument calibration constants ($K_{ray}$, $K_{mie}$), the
channel cross-talk coefficients $C_1$, $C_2$, $C_3$ and $C_4$, the laser pulse energy ($E_0$) and the contributions
from the pure molecular (X) and particulate (Y) signals (see Equations 1 and 2 in Flament et al.
(2021)). The latter ones, for each bin, result from the vertical integration of the backscatter (either
molecular or particulate) where the squared one-way transmission through the atmosphere is taken
into account (see Equations 3 and 4 in Flament et al. (2021)).
The separation of the molecular and particle signals on each channel is imperfect, due to the
HSRL instrument design, which makes necessary a cross-talk correction. The channel cross-talk
corresponding to the transmission of the Rayleigh-Brillouin spectrum (depending on the temperature,
pressure and the Doppler shift) through the Rayleigh and Mie channels is expressed by the calibration
coefficients $C_1$ and $C_4$, respectively (Flament et al., 2021). The other two coefficients, $C_2$ and $C_3$,
refer to the transmission of a Mie spectrum (depending on the Doppler shift) through the Mie and
Rayleigh channels, respectively. Along with the "C coefficients", the instrument calibration constants
($K_{ray}$, $K_{mie}$) (see in Flament et al., 2021) are included in the AUX_CAL files.
Finally, the cross-talk corrected signals, normalized with the range bin thickness and corrected
by the range between the satellite and the observed target, are utilized for the retrieval of the vertically
resolved backscatter ($\beta$) and extinction ($\alpha$) coefficients. The former, at each bin, is derived by the Y/X
ratio multiplied with the molecular backscatter coefficient (see Equations 9 and 10 in Flament et al.,



2021) computed from the pressure and temperature ECMWF simulated fields according to Collis and
Russel (1976). For the L2A extinction retrievals, derived via an iterative process from top to bottom,
a method called normalized integrated two-way transmission (NITWT) is applied, using measured
and simulated pure molecular signals, under the assumption that the particles' extinction at the top-
most bin is zero (see equations 11-14 in Flament et al., 2021). This consideration makes the
"downwards" solution of the integral equations quite sensitive to the noise within the topmost bin (at
altitudes ~20-25 km), which is used as reference for the normalization, particularly under low SNR
conditions due to the low molecular density. This is a challenge frequently faced for the Aeolus
observations due to the weaker measured signals than those of the pre-launch expectations (Reitebuch
et al., 2020) as well as to the possible presence of stratospheric aerosols within the top-most range
bin or above. In principle, the extinction is retrieved recursively taking into account the attenuation
from the overlying bins and by contrasting observed and simulated molecular signals. By
differentiating two consecutive bins, unrealistically high positive or negative extinctions can be
retrieved (see Fig. 10 in Flament et al., (2021)) resulting from fluctuations between strong and weak
attenuation. In the case of negative extinctions, the SCA algorithm regularizes the solution by
resetting extinction to zero (Ehlers et al., 2021), which can lead to an underestimation of the partial
column transmission. In order to compensate the impacts of the aforementioned issues, it has been
shown by error propagation calculations (see equations 18 and 19 in Flament et al. (2021)), that by
averaging two consecutive bins the retrieved extinction becomes more reliable at the expense of the
vertical resolution (23 bins; "mid-bin" vertical scale). In contrast to SCA, in the SCA mid-bin
negative extinctions can be found since the zero-flooring constraint is not implemented. For
consistency reasons, the averaging between two neighboring bins is applied also in the backscatter
coefficient thus allowing the derivation of the lidar ratio.

**4. Ground-based lidars (PANACEA)**
The ground-based observational datasets used herein, are taken from stations that participate
in the PANhellenic infrastructure for Atmospheric Composition and climatE chAnge (PANACEA)
initiative. Within PANACEA, adverse measurement techniques and sensors are utilized in a
synergistic way for monitoring the atmospheric composition and climate change related parameters
in Greece.
The locations of the stations providing routine measurements to the PANACEA network are
shown in Figure 1-i. For the assessment analysis of Aeolus L2A products, we utilize available
measurements from PANACEA stations, namely Antikythera (ANT), Athens (ATH) and
Thessaloniki (THE), equipped with multiwavelength polarization lidar systems. All stations comply



with the quality-assurance criteria established within EARLINET (e.g. see Freudenthaler et al., 2016)
so as to assure the provision of high-quality aerosol related products. Consequently, the derived
datasets can be considered for any validation purpose. To ensure the homogeneity of the optical
property profiles derived from the adverse lidar systems operating in each station, the Single Calculus
Chain algorithm (SCC; D' Amico et al., 2016; Mattis et al., 2016) was used; an automatic processing
chain for lidar data, developed within EARLINET. All systems employ multiple detectors, operating
either in the photon-counting or analog mode. Herein elastically and inelastically backscattered
signals at 355 and 387 nm, were used to evaluate Aeolus products. The optical property profiles were
derived using the Raman and Klett-Fernald-Sassano inversion methods (Ansmann et al. 1992;
Fernald, 1984; Klett, 1981; Sasano and Nakame, 1984) during night-time and daytime measurements
respectively. Below the full overlap height and under the assumption of a well-mixed boundary layer,
lidar profiles can be linearly extended to the ground (Siomos et al., 2018, Baars et al., 2016).

*4.1 Antikythera*
Regular lidar measurements have been performed at the PANGEA observatory (PANhellenic
GEophysical observatory of Antikythera; lat=35.86 N, lon=23.31 E, alt=193 m asl.) contributing to
this study. Under the prevailing Mediterranean background conditions, and being across the traveled
path of different air masses (i.e. marine particles, Saharan dust), Antikythera is considered as an ideal
location for Cal/Val activities.
The lidar system deployed at PANGEA is operated by the National Observatory of Athens
(NOA). It is a Polly[XT] (Engelmann et al., 2016) multi-wavelength Polarization-Raman-Water vapor
lidar, designed for unattended, continuous operation. Polly [XT] deploys an Nd:YAG laser which emits
linearly polarized light at 355, 532 and 1064 nm. The radiation elastically and inelastically
backscattered from aerosol, cloud particles, nitrogen (at 387 and 607 nm) and water vapor (at 407
nm) molecules, is collected using a near-range (spherical mirror of 50 mm diameter, focal length
f=250 mm and 2.2 mrad field of view (FOV)) and a far-range receiver (Newtonian telescope with a
300 mm diameter primary mirror, f=900 m and 1 mrad FOV) at a raw vertical resolution of 7.5m.
The combined use of the near-range and far-range receivers allows for the retrieval of the aerosol
optical properties from 500 m up to ~12-14 km above the ground. A detailed description of the
technical characteristics of Polly[XT] can be found in Engelmann et al. (2016).

*4.2 Athens*
The Laser Remote Sensing Unit of the National and Technical University of Athens, Greece
(LRSU; NTUA; lat=37.96 N, lon=23.78 E, alt=200 m asl.), is part of the EARLINET since May



2000. Currently, the Athens lidar station performs simultaneous measurements with two different
lidar systems, EOLE and DEPOLE. The EOLE lidar is an advanced 6-wavelength elastic
backscatter/Raman lidar system able to provide the aerosol backscatter coefficient at 355, 532 and
1064 nm, the aerosol extinction coefficient at 354 and 532 nm and water vapor mixing ratio profiles
in the troposphere. EOLE is based on a pulsed Nd:YAG laser system and a 300 mm diameter
Cassegrain telescope (f=600 mm, FOV =1.5 mrad) which collects all elastically backscattered lidar
signals (355-532-1064 nm), as well as those generated by the spontaneous Raman effect (by
atmospheric $N_2$ at 387-607 nm and by $H_2O$ at 407 nm). The full overlap (i.e. the altitude from which
upwards the whole lidar beam is within the telescope FOV) of EOLE is reached at, approximately,
812 m a.s.l..

314   The DEPOLE lidar is a depolarization lidar, able to provide profiles of the aerosol backscatter

coefficient and the linear particle/volume depolarization ratio at 355 nm. DEPOLE is based on a
pulsed Nd:YAG laser system which emits linearly polarized light at 355 nm. The elastically
backscattered lidar signal at 355 nm is collected by a 200 m diameter Dall-Kirkham/Cassegrain
telescope (f=600 mm, FOV=3.13 mrad) and the full overlap is reached at, approximately, 500 m a.s.l..

*4.3 Thessaloniki*

321   Thessaloniki's multiwavelength Polarization Raman lidar system (THELISYS) belongs to the

Laboratory of Atmospheric Physics that is located in the Physics Department of the Aristotle
University of Thessaloniki (lat = 40.63 N, lon = 22.96 E, a.s.l. = 50m). Thessaloniki is a member
station of the EARLINET since 2000, providing almost continuous measurements, according to the
network schedule (every Monday morning, ideally close to 12:00 UTC, and every Monday and
Thursday evening) and during extreme events (e.g., Saharan dust outbreaks, smoke transport from
biomass burning, volcanic eruptions) and satellite overpasses. THELISYS has been validated within
EARLINET at hardware level by two intercomparison campaigns (Matthias et al., 2004), in order to
fulfill the standardized criteria. The system is based on the first (1064 nm), second (532 nm), and
third harmonic (355 nm) frequency of a compact, pulsed Nd:YAG laser emitted with a 10 Hz
repetition rate. THELISIS setup includes three elastic backscatter channels at 355, 532 and 1064nm,
two nitrogen Raman channels at 387 nm and 607nm, and two polarization sensitive channels at 532
nm. The acquisition system is based on a LICEL Transient Digitizer working in both the analogue
and photon counting (250 MHz) mode. The vertical resolution of the elastic raw signal at 355 nm is
equal to 3.75 m and is recorded in both analog and photon counting mode. The full overlap height is
almost 800m a.s.l. A detailed description of THELISIS can be found in Siomos et al. (2018) and
Voudouri et al. (2020).





*4.4. Aerosols' load variability in the vicinity of the PANACEA sites*


The variability of the atmospheric aerosol load in the vicinity of three PANACEA stations
(Fig. 1-i) is discussed in this section. The aim of this introductory analysis is to investigate the
horizontal homogeneity of the aerosol optical depth (AOD) in the respective broader areas, playing a
key role in the comparison of ground-based and spaceborne profiles, which are not spatially
coincident as it will be shown in a following section (i.e., collocation method). For the purposes of
this analysis, we have processed the mid-visible (550 nm) AOD retrievals, over the period 2008-2017,
acquired by the MODIS sensor, mounted on the Aqua polar orbiting satellite. More specifically, we
have analyzed the Level 2 (L2) MODIS-Aqua AODs, obtained by the latest version (Collection 6.1)
of the operational retrieval algorithms (Remer et al., 2008; Levy et al., 2013; Sayer et al., 2013),
accessible from the Level 1 and Atmosphere Archive and Distribution System (LAADS) Distributed
Active Archive Center (DAAC) (https://ladsweb.modaps.eosdis.nasa.gov/, last access: 17 June
2022).

For each station, we have calculated the average AOD values within progressively larger
circular areas, with radii spanning from 10 to 100 km with an incremental step of 10 km (Fig. 1-ii).
Figure 1-iii illustrates the resulting AODs for each station (x labels) and at each radius (colored bars).
In order to ensure the reliability of the obtained results, only the best (QA=3) MODIS-Aqua AOD L2
retrievals are considered whereas the spatial averages are calculated only when the satellite
observations are simultaneously available at all circles. In the urban areas of Athens (ATH) and
Thessaloniki (THE), the contribution of anthropogenic aerosols on the columnar load fades for
increasing radii. On the contrary, at Antikythera (ANT), the spatial AOD means remain almost
constant revealing a horizontal homogeneity of the aerosol load in the broader area. An alternative
way to compare the differences in the AOD spatial representativeness between the urban (ATH, THE)
and the remote (ANT) sites is depicted in Fig. 1-iv showing the normalized values for each radius
with respect to the AOD levels of the inner circle (i.e., up to 10 km distance from the station). In both
urban sites the values are lower than one (dashed line), decreasing steadily in THE and smoothly in
ATH after an abrupt reduction from 10 to 20 km. In ANT, the blue curve resides almost on top of the
dashed line, throughout the circles radii (i.e., range of distances) indicating the absence of significant
horizontal variation of the aerosol load suspended in the surrounding area of the station.

**5. Collocation between Aeolus and ground-based lidars**

The assessment of Aeolus L2A backscatter profiles has been performed against the
corresponding measurements acquired at the three EARLINET/PANACEA lidar stations. In Figure
2, three examples of the collocation between ground-based and spaceborne retrievals are illustrated





in order to describe our approach as well as to clarify points needed in the discussion of the evaluation
results (Section 6). At each station, we identify the observations (BRCs), considering their
coordinates at the beginning of the ALADIN scan, falling within a circle of 120 km radius (black
dashed circle) centered at the station coordinates (black dot). Based on the defined spatial criterion,
the number of BRCs residing within the 120 km circle should be at least one and cannot be more than
three. We denote each one of them, along the ALADIN measurement track (white stripe), with
different colors (red, blue and magenta) in Fig. 2. The orange arrow shows the flight direction of the
satellite for the dusk (ascending) or dawn (descending) orbits. For the ground-based observations, the
aerosol backscatter profiles are derived considering a time window of ± 1 hour around the satellite
overpass. Nevertheless, this temporal collocation criterion has been relaxed or shifted in few cases to
improve the quality of the ground-based retrievals as well as to increase the matched pairs with Aeolus
L2A profiles. Overall, 46 cases are analyzed out of which 15 have been identified over Antikythera,
12 in Athens and the rest 16 in Thessaloniki.
The ground-based profiles are derived under cloud free conditions in contrast to Aeolus L2A
backscatter profiles providing aerosol and/or cloud backscatter. Therefore, a cloud screening of the
Aeolus data using auxiliary cloud information was applied. In the framework of the present study, the
exclusion of cloud contaminated Aeolus profiles relies on the joint processing of the cloud mask
product (CLM; https://www.eumetsat.int/media/38993) derived by radiances acquired by the SEVIRI
(Spinning Enhanced Visible and Infrared Imager) instrument mounted on the Meteosat Second
Generation (MSG4) geostationary satellite (Schmetz et al., 2002). It should be noted, however, that
the CLM product serves as an indication of clouds presence, without providing information about
their macrophysical properties (i.e., cloud coverage), their phase (i.e., ice, water, mixed) or their
categories (i.e., low, middle, high). In the illustration examples of Figure 2, the grey shaded areas
represent the spatial coverage of CLM in the broader area at each PANACEA site. Based on the
filtering procedures, the Aeolus L2A backscatter retrievals, throughout the probed atmosphere by
ALADIN, are removed from the analysis when the grey shaded areas overlap with a BRC.
**6. Results**
*6.1 Assessment of Aeolus L2A backscatter under different aerosol scenarios*
In the first part of the analysis we assess the quality of the Aeolus L2A backscatter under
various aerosol regimes aiming to: (i) investigate the capabilities of the ALADIN spaceborne lidar to
detect aerosol layers, (ii) investigate how the horizontal homogeneity and vertical structure of the
aerosol layers can affect the level of agreement between spaceborne and ground-based retrievals and
(iii) demonstrate the synergistic use of various datasets for a better characterization of the prevailing



aerosol conditions. All of these aspects are necessary towards a comprehensive Cal/Val study for
facilitating the interpretation of the obtained findings and at a further step for identifying possible
upgrades on Aeolus retrievals. Overall, four cases over the Antikythera island (southwest Greece) are
analyzed for the Aeolus L2A aerosol backscatter retrievals (Baseline 2A11) and the obtained results
are depicted in Figure 3.

As it has been already mentioned, Aeolus retrievals are representative at coarse spatial (BRC

level; ~90 km) and vertical (minimum 250 m) resolution, while currently there is no scene
classification scheme. In order to overcome this inherent limitation, as much as possible, several
ancillary data and products are utilized in parallel with those of the MSG-SEVIRI CLM product.
Based on the FLEXPART v10.4 Lagrangian transport model (Stohl et al., 2005; Ignacio Pisso et al.,
2019) we have reproduced the 5-day air masses backtrajectories prior to their arrival at 7 altitudes
above the ground station. FLEXPART was driven with 3-hourly meteorological data from the
National Centers for Environmental Prediction (NCEP) Global Forecast System (GFS) analyses
provided at 0.5° × 0.5° resolution and for 41 model sigma pressure levels
(https://nomads.ncep.noaa.gov/txt_descriptions/GFS_half_degree_doc.shtml). For depicting the
spatial patterns of the mid-visible (550 nm) total and speciated AOD, we are relying on the MERRA-
2 (Modern-Era Retrospective analysis for Research and Applications version 2; Buchard et al., 2017;
Randles et al., 2017; Gelaro et al., 2017) and CAMS (Copernicus Atmosphere Monitoring Service;
Inness et al., 2019) reanalysis datasets, both providing aerosol products of high quality (Gueymard
and Yang, 2020; Errera et al., 2021). Finally, AERONET sun-direct measurements (Level 1.5,
Version 3; Giles et al., 2019; Sinyuk et al., 2020) of spectral AODs and Ångström exponent are also
used for the characterization and the temporal evolution of the aerosol load over the station.

*6.1.1 Dust advection on 10$^{th}$ of July 2019*

The first case refers to the advection of dust aerosols from northwest Africa towards the

Antikythera island with dust-laden air masses crossing southern Italy prior to their arrival to the
PANACEA site from northwest directions (Figure S4). This route of air masses, driven by the
prevailing atmospheric circulation (Gkikas et al., 2015), is typical during summer when Saharan
aerosols are advected towards the eastern Mediterranean (Balis et al., 2006). MERRA-2 (Fig. S3-i)
and CAMS (Fig. S3-ii) show a reduction of AODs from west to east whereas the large contribution
(>80%) of dust aerosols to the total aerosol load is evident in both reanalysis products (results not
shown here). The moderate-to-high AOD values are confirmed by the ground-based sunphotometric
measurements (Fig. S1) and are associated with low Ångström exponent values (0.2 – 0.4) thus
indicating the prevalence of coarse particles. This is further supported from Polly$^{XT}$ measurements



(Fig. S2) revealing persistent dust layers associated with volume linear depolarization ratio (VLDR)
values of 5-10% at 355 nm, stretched from altitudes close to the ground and up to almost 6 km.
This case is ideal for evaluating L2A backscatter retrievals since non-spherical mineral
particles are probed by ALADIN, which does not detect the cross-polar component of the
backscattered lidar signal. Therefore, a degradation of ALADIN's performance is expected (i.e.,
underestimation of the backscatter coefficient and overestimation of the lidar ratio) when aspherical
particles (e.g., dust, volcanic ash, cirrus ice crystals) are probed. In Figure 3, the backscatter
coefficient step-like vertical profiles from Aeolus at the regular (brown) and mid-bin (black) vertical
scales are compared against those acquired by the Polly$^{XT}$ (pink) at 355 nm. The colored dashed lines
(Aeolus) and the pink shaded area (Polly$^{XT}$) correspond to the statistical uncertainty margins of the
spaceborne and the ground-based (D'Amico et al., 2016) retrievals, respectively. At a first glance, it
is revealed that the geometrical structure of the dust layer, extending from 1 to 6 km, is generally well
captured by ALADIN (except at altitude ranges from 1 to 2.5 km), but the backscatter magnitude is
constantly underestimated. A fairer comparison, considering that depolarizing particles are recorded,
requires the conversion of the backscatter retrievals assuming that Polly$^{XT}$ emits circularly polarized
radiation (instead of linearly polarized) and thus resembling ALADIN. Under the assumption of
randomly oriented particles and negligible multiple scattering effects, this transformation is made
based on theoretical formulas (Mishchenko and Hovenier, 1995; Roy and Roy, 2008), as it has been
shown in Paschou et al. (2021). Following this approach, the Aeolus-like backscatter (i.e., circular
co-polar component; blue curve in Fig. 3) is reproduced for the ground-based profiles at altitudes
where UV depolarization measurements are available. Thanks to this conversion, the Aeolus-Polly$^{XT}$
negative biases diminish and the Aeolus-like curve resides closer to those of SCA (brown) and SCA
mid-bin (black) backscatter levels. The difference between pink and blue backscatter profiles, ranging
from 13 to 33% in this specific case, reflects the underdetermination of the particle backscatter
coefficient in case of depolarizing aerosols being probed, due to the missing cross-polar backscatter
component.

*6.1.2 Long-range transport of anthropogenic aerosols on 3$^{rd}$ July 2019*
Under the prevalence of the Etesian winds (Tyrlis and Lelieveld, 2013), a typical pattern
dominating over the broader Greek area during summer months, when winds blow mainly from N-
NE directions, anthropogenic aerosols from megacities (Kanakidou et al., 2011) and particles
originating from biomass burning in the eastern Europe and in the surrounding area of the Black Sea
(van der Werf et al., 2017) are transported southwards. Based on the FLEXPART simulations (Fig.
S8), the air masses carrying fine particles, gradually descend till their arrival over Antikythera from



north-northeastern directions. During early morning hours, when ALADIN probes the atmosphere at
a distance of ~90 km westwards of the ground station (dawn orbit; descending), moderate AODs (up
to 0.15 at 340 nm) and very high Ångström exponent values (>1.2) are measured with the Cimel
sunphotometer (Fig. S5). The aerosol load is confined below 2.5 km consisting of spherical particles
as it is revealed from the Polly$^{XT}$ volume linear depolarization ratio (VLDR) values, which do not
exceed 5% at 355 nm (Fig. S6). In the vicinity of the PANGEA observatory, MERRA-2 (Fig. S7-i)
and CAMS (Fig. S7-ii) AODs, mainly attributed to organic carbon, sulphate and sea-salt aerosols, do
not exceed 0.2 and they are coherent in spatial terms (i.e., horizontal homogeneity). In this case,
Polly$^{XT}$ particle backscatter coefficient profiles coincide with the corresponding Aeolus-like profiles
(blue and pink curves are almost overlaid in Fig. 3-ii) since depolarization values are negligible.
Under these conditions, ALADIN is capable of reproducing satisfactorily the layer's structure (SCA
retrievals - brown curve) whereas slightly overestimates its intensity (SCA mid-bin retrievals - black
curve) with respect to the ground-truth retrievals.

*6.1.3 Long range transport of fine aerosols on 8$^{th}$ July 2020*
On 8$^{th}$ July 2020, the broader area of the Antikythera island was under the impact of moderate-
to-high aerosol loads, mainly consisting of organic and sulphate particles based on CAMS simulated
AODs (up to 0.5) in the western and southern sector of the station (Fig. S11-ii). The prevalence of
fine aerosols is confirmed by the AERONET measurements, yielding UV AODs up to 0.5 and
Ångström exponent higher than 1.5 during early afternoon (Fig. S9). MERRA-2 AOD patterns (Fig.
S11-i) and speciation (strong contribution from marine and sulphate aerosols to the total aerosol load)
are different from those of CAMS, indicating a moderate performance with respect to the ground-
based sunphotometer observations (Fig. S9). Air masses originating in northern Balkans and the
Black Sea, after crossing metropolitan areas (i.e., Istanbul, Athens), are advected over ANT at
altitudes up to 4 km above surface, whereas a second cluster aloft (>5 km) indicates the convergence
of air masses from northwest (Fig. S12). In vertical terms, aerosol layers with local backscatter
maxima gradually reducing from 3.5 to 1.5 Mm$^{-1}$ sr$^{-1}$ are observed up to 4 km based on Polly$^{XT}$
backscatter coefficient profiles (pink curve, Fig. 3-iii) whereas almost identical values are recorded
for the Aeolus-like retrievals (blue curve, Fig. 3-iii). Aeolus performance reveals an altitude
dependency according to the comparison versus Polly$^{XT}$ vertically resolved retrievals. From top to
bottom, the weak layer, extending from 6 to 8 km, observed in the ground-based lidar profiles is
partially evident in the Aeolus retrievals. At height ranges (< 4 km) where the main portion of the
aerosol burden resides, there is a contradiction of ALADIN's performance clearly seen beneath and
above ~2 km. In the free troposphere, the retrieved backscatter by ALADIN is underestimated with





respect to Polly$^{XT}$ retrievals whereas the position of the Aeolus local maximum backscatter (~1 Mm$^-$
$^1$ sr$^{-1}$) is recorded exactly above the top of the aerosol layer observed from the ground. On the contrary,
below 2 km, the agreement between ALADIN and Polly$^{XT}$ becomes better, particularly for SCA mid-
bin, even though the narrow peak recorded at ~1.2 km by Polly$^{XT}$ cannot be reproduced by ALADIN.
This might be attributed either to the adjusted RBS at the lowermost bin (1 km thickness) or to the
lower accuracy of Aeolus retrievals near the ground due to the attenuation from the overlying layers
(Flament et al., 2021).

*6.1.4 Stratification of spherical and non-spherical particles on 5$^{th}$ August 2020*

In the last case, that took place on 5$^{th}$ August 2020, we are investigating the ability of Aeolus

to reproduce adequately the vertical structure of an aerosol layer detected up to 4 km based on Polly$^{XT}$
(Fig. 3-iv; pink curve). The "peculiarity" of this study case, as it is revealed by the Polly$^{XT}$ time-
height plots of VLDR (Fig. S14), is that spherical fine particles dominate below 2.5 km whereas the
presence of non-spherical coarse aerosols above this layer is evident. This stratification results from
the convergence of air masses either originating in central Europe or suspending most of their travel
above northwest Africa (Fig. S12). According to MERRA-2 (Fig. S11-i) and CAMS (Fig. S11-ii)
reanalysis datasets, AODs fade from west to east while both numerical products indicate the
coexistence of carbonaceous, sulphate and mineral particles over the area where ALADIN samples
the atmosphere (~100 km westwards of Antikythera). During the Aeolus overpass (~04:40 UTC),
sunphotometer columnar observations are not available (Fig. S13). However, one hour later, UV
AODs up to 0.4 are recorded and remain relatively constant during sunlight hours. At the same time,
intermediate Ångström values (0.7 – 1), exhibiting weak temporal variation, indicate a mixing state
of fine and coarse aerosols. In the lowest troposphere (< 2km), Aeolus overestimates significantly the
backscatter coefficient but reproduces satisfactorily the aerosol layer structure at the mid-bin vertical
scale (i.e., SCA mid-bin; black curve; Fig. 3-iv), in contrast to the regular scale (i.e., SCA; brown
curve; Fig. 3-iv). It is reminded that SCA backscatter is actually retrieved whereas the SCA mid-bin
results by averaging two consecutive bins following the procedure applied on the extinction for
mitigating the downwards error propagation in the retrieval algorithm solution (Flament et al., 2021).
At higher altitudes (2.5 – 4 km), due to the suspension of depolarizing mineral particles, a declination
is marked between the pink (linear-derived) and blue (Aeolus-like) Polly$^{XT}$ profiles. Again, the SCA
mid-bin backscatter performs better than those of SCA reproducing more realistically the shape and
the magnitude of the Polly$^{XT}$ Aeolus-like profile. Finally, ALADIN detects aerosol layers between
5.5 and 8 km, assuming that clear-sky conditions are appropriately represented in the MSG-SEVIRI
imagery and remain constant within the time interval (~6 minutes) of MSG and Aeolus observations,
and the SCA mid-bin backscatter resides closer to the Polly[XT] levels, which, however, are noisy.

A general remark that should be made, is that for the cases analyzed, between the ground-

based and spaceborne profiles there is an inconsistency in the vertical representativeness within the
lowermost Aeolus bin. Under the absence of the near-field receivers (not considered in our study)
Polly[XT] profiles are reported above ~800 m where the overlap between the laser beam and the receiver
telescope field of view is expected to be full. However, the base altitude of the near-surface Aeolus
bin is at ~200 m. This can interpret, at some degree, the large positive ALADIN-Polly[XT] departures
at altitudes below 1 km, which are possibly further strengthened by an inappropriate RBS (i.e., low
SNR) in the Aeolus retrievals.

*6.2 Overall assessment and dependencies*

In the second part of the analysis, an overall assessment of the Aeolus L2A retrievals is

performed. Due to the very limited availability of ground-based extinction profiles, only the Aeolus
L2A backscatter observations are evaluated. It must be clarified that the evaluation of the Aeolus
satellite (SAT) backscatter coefficient is conducted without any conversion (i.e., from total linear to
circular co-polar) of the ground-based lidar (GRD) profiles. This has been decided since many of the
SAT-GRD collocated samples are derived from the Thessaloniki station. Due to technical issues
(related to the polarization purity of the emitted laser beam and the performance of the telescope
lenses) no calibrated depolarizing measurements, necessary to derive the Aeolus-like products
(Paschou et al., 2021), are available for the study period. Nevertheless, we are not expecting that this
consideration, acknowledging that it is imperfect, will affect substantially the robustness of our
findings since in most of the study cases the contribution of depolarizing particles is quite low based
on the ancillary datasets/products. The discussion in the current section is divided in two parts. First,
the vertically resolved evaluation metrics are presented separately for the two Aeolus vertical scales,
both for the unfiltered and the filtered (cloud-free) profiles (Section 6.2.1). The same analysis format
(i.e., SCA vs SCA mid-bin, unfiltered vs filtered) is kept in the second sub-section (Section 6.2.2)
where the evaluation results are presented as a function of various dependencies.

*6.2.1 Vertically resolved evaluation metrics*

In Figure 4, the vertically resolved bias (SAT-GRD; upper panel) and root mean square error

(RMSE; bottom panel) metrics are depicted for the unfiltered (cloud and aerosol backscatter) Aeolus
L2A backscatter retrievals, reported at the regular (left column) and the mid-bin (right column)
vertical scales. Bias and RMSE metrics are used in a complementary way in order to avoid any



misleading interpretation of the former score attributed to counterbalancing negative and positive
SAT-GRD deviations. For the calculation of the evaluation scores, the GRD profiles have been
rescaled to match Aeolus vertical product resolution. Note that in the SAT-GRD pairs, all BRCs from
all cases are included (right y-axis in Figure 4), satisfying the defined collocation criteria (see Section
5), and they are treated individually. Aeolus L2A data are provided vertically at a constant number of
range bins (i.e., 24 for SCA and 23 for SCA mid-bin) but their base altitude and their range vary along
the orbit and from orbit-to-orbit and they are defined dynamically (depending on the optimum SNR).
Therefore, since the GRD and SAT profiles are not interpolated in a common predefined grid, we are
using as reference the reverse index (with respect to those considered in the SCA retrieval algorithm
in which 1 corresponds to the top-most bin) of Aeolus SCA (from 1 to 24; left y-axis in Figs 4 i-a and
ii-a) and SCA mid-bin (from 1 to 23; left y-axis in Figs 4 i-b and ii-b) vertical scales.

According to our results for the unfiltered Aeolus backscatter profiles (Fig. 5), positive biases

(up to 3.5 Mm$^{-1}$ sr$^{-1}$; red bars) are evident, at both vertical scales, at the first three bins (below 2 km).
For altitude ranges spanning from 2 to 8 km (bins 4 – 12), mainly positive SAT-GRD biases (up to
~1.5 Mm$^{-1}$ sr$^{-1}$) are recorded for SCA mid-bin whereas for SCA reach up to ~1 Mm$^{-1}$ sr$^{-1}$ in absolute
terms. Similar tendencies are evident at the highest altitudes (> 8 km) but the magnitude of the SAT-
GRD offsets becomes lower (< 0.5 Mm$^{-1}$ sr$^{-1}$). Between the two Aeolus vertical scales, SCA mid-bin
performs better than SCA up to ~8 km (bin 12) and similar aloft, as it is shown by the RMSE profiles
(bottom panel in Fig. 4). Nevertheless, the most important finding is that Aeolus is not capable to
reproduce satisfactorily the backscatter profiles as it is revealed by the RMSE levels, which are
maximized near the ground (~ 8 Mm$^{-1}$ sr$^{-1}$), are considerably high (up to 6 Mm$^{-1}$ sr$^{-1}$) in the free
troposphere and are minimized (< 1 Mm$^{-1}$ sr$^{-1}$) at the uppermost bins. Our findings are highly
consistent with those presented in Abril-Gago et al. (2022), who performed a validation of Aeolus
L2A particle backscatter coefficient against reference measurements obtained at three
ACTRIS/EARLINET sites in the Iberian Peninsula. Several factors contribute to the obtained height-
dependen SAT-GRD discrepancies. Near the ground, the observed maximum overestimations are
mainly attributed to the: (i) contamination of the ALADIN lidar signal by surface reflectance, (ii)
increased noise in the lowermost bins and (iii) limited vertical representativeness of the GRD profiles
below 1 km. On the contrary, in the free troposphere, the cloud contamination on spaceborne
retrievals plays a dominant role on the occurrence of ALADIN backscatter overestimations with
respect to the cloud-free ground-based retrievals. From a statistical point of view, it must also be
mentioned that the robustness of the bias and RMSE metrics decreases for the increasing altitudes
due to the reduction of the number of the SAT-GRD matchups (right y-axis in Fig. 4) participating in
the calculations.





The assessment analysis has been repeated after removing Aeolus profiles when clouds are
detected by MSG-SEVIRI (grey shaded areas in Fig. 1) within a BRC (colored rectangles in Fig. 1).
By contrasting Figures 4 and 5 (evaluation metrics for the filtered profiles), an expected improvement
of the level of agreement between SAT and GRD is visible. This translates into a drastic reduction of
bias and RMSE values at altitude ranges up to 5-6 km (~bin 12). Between bins 2 and 5 slight
underestimations (blue bars) and overestimations (red bars) are found for SCA (Fig. 5 i-a) whereas
low positive SAT-GRD offsets are recorded for SCA mid-bin (Fig. 5 i-b). Above bin 5, SAT-GRD
deviations are low in absolute terms, oscillating around zero, for SCA, whereas only positive SAT-
GRD biases are recorded for SCA mid-bin, which are maximized (~ 0.7 Mm$^{-1}$ sr$^{-1}$) at the highest bins
and are associated with limited SAT-GRD matchups (right x-axis in Fig. 5 i-b). The obtained
improvements on bias scores become more confident since they are associated with similar strong
reductive tendencies on RMSE levels. More specifically, the RMSE spikes of extremely high values
recorded in the unfiltered profiles either disappear or weaken in the case of the Aeolus filtered SCA
(Fig. 5 ii-a) and SCA mid-bin (Fig. 5 ii-b) backscatter profiles. However, even though the RMSE
values at the lowermost bins are decreased when cloud contaminated Aeolus profiles are eliminated,
still the corresponding levels for the filtered profiles are considerably high attributed to the higher
SNR and the possible impact of surface returns.

*6.2.2 Scatterplots*

An alternative approach to assess the performance of Aeolus L2A backscatter is attempted
here by reproducing two dimensional histograms for the entire SAT-GRD collocated sample as well
as scatterplots resolved based on various dependencies, aiming to investigate the factors determining
the level of agreement between spaceborne and ground-based retrievals. More specifically, the
dependencies under investigation are those of the: (i) station locations, (ii) BRCs and (iii) orbits (dawn
vs dusk). The evaluation metrics have been calculated for all possible combinations of vertical scales
(SCA vs SCA mid-bin) and Aeolus profiles (unfiltered vs filtered).
Figure 6 depicts the two-dimensional histograms between GRD (x-axis) and SAT (y-axis)
backscatter coefficient for the raw (upper panel) and filtered (bottom panel) Aeolus profiles reported
at the SCA (left column) and SCA mid-bin (right column) vertical scales. Note that we have removed
SAT-GRD pairs in which Aeolus backscatter exceeds 20 Mm$^{-1}$ sr$^{-1}$ in order to avoid the
"contamination" of extreme outliers in the calculated metrics, possibly attributed to the presence of
clouds (Proestakis et al., 2019). It is also clarified that the Aeolus QA flags are not taken into account
in the current study, since their validity is not yet reliable (Reitebuch et al., 2020) as it has been
demonstrated in Abril-Gago et al. (2022).



Between SCA and SCA mid-bin unfiltered retrievals there is a contradiction of which
performs better relying on the correlation coefficient (0.36 and 0.39, respectively), bias (0.45 and
0.69, respectively) and RMSE (2.00 and 1.88, respectively) metrics. After removing cloud-
contaminated Aeolus profiles, the amount of the SAT-GRD matchups is reduced by about 55% and
59% for SCA (from 537 to 239) and SCA mid-bin (from 356 to 147), respectively. Nevertheless,
thanks to this filtering procedure, the initially observed overestimations for SCA and SCA mid-bin
are reduced by ~25% and ~43%, respectively, whereas the RMSE values drop down to 1.65 (SCA)
and 1.00 (SCA mid-bin). The better agreement between SAT and GRD, for the filtered Aeolus
profiles, is further justified by the increase of the R values (from 0.39 to 0.48) for the SCA mid-bin
whereas for SCA there is not any positive or negative tendency (R=0.36). The spread of the points in
the two dimensional space reveals many similarities with the corresponding scatterplots presented in
Abril-Gago et al. (2022) for the Iberian ACTRIS/EARLINET stations.
A common feature in all scatterplots, shown in Figure 6, is that most of the positive outliers
are found at the lowermost bins (see Figs. 4 and 5). SAT beta can reach up to 20 Mm$^{-1}$ sr$^{-1}$ in contrast
to the corresponding GRD levels, which are mainly lower than 2 Mm$^{-1}$ sr$^{-1}$. For SCA (Figs. 6 i-a, 6
ii-a), the majority of the negative SAT-GRD pairs are recorded at the highest bins in which, however,
both spaceborne and ground-based backscatter coefficients are noisy. Another cluster of SAT-GRD
pairs is those where slight negative Aeolus backscatter values are grouped together with low positive
backscatter values retrieved from ground. At the mid-bin vertical scale, for the unfiltered Aeolus
profiles (Fig. 6 i-b), the negative SAT backscatter values are masked out resulting in better evaluation
metrics (except the increase of bias due to the removal of the negative Aeolus backscatter) with
respect to the regular vertical scale. Among the four scatterplots, the best agreement between Aeolus
and ground-based retrievals is revealed for the SCA mid-bin filtered profiles (Fig. 6 ii-b) attributed
to the coincident elimination of the negative and the extreme positive Aeolus backscatter coefficient.
Figure 7 depicts the overall scatterplot between ground-based and spaceborne retrievals as a
function of the three PANACEA sites (colored categories). The associated evaluation scores are
summarized in Table 1 and 2 for the unfiltered and filtered Aeolus profiles, respectively. The majority
of the extreme positive outliers of unfiltered SCA retrievals (Fig. 7 i-a) are recorded in Thessaloniki
and Athens. According to our results, significant overestimations (0.73 for ATH and 0.83 for THE)
and high RMSE values (2.26 for ATH and 2.60 for THE) are found. At Antikythera island (ANT),
the biases are quite low and equal to 0.06 and 13.6% in absolute and relative terms, respectively
(Table 1). In all stations, for the unfiltered SCA mid-bin retrievals, the absolute SAT-GRD departures
become larger whereas the RMSE decreases in ANT/THE and increases in ATH. Regarding the
temporal covariation between SAT and GRD retrievals, a noticeable improvement is evident in ANT



(i.e., R increases from 0.49 to 0.57). For the quality-assured Aeolus profiles (Table 2), all evaluation
metrics converge towards the ideal scores for SCA mid-bin whereas mainly positive tendencies (i.e.,
better agreement) are evident for SCA. Overall, among the three stations the best performance of
Aeolus is recorded at Antikythera island.
Between dawn (descending) and dusk (ascending) orbits, better bias and RMSE scores are
computed when Aeolus is flying during early morning hours while better R values are found during
early afternoon satellite overpasses. However, our orbit-based results are not robust since the number
of Aeolus overpasses is not evenly distributed (about 85% of the SAT-GRD matchups are acquired
during dawn orbits). Among the three BRCs (red, blue or magenta), which can satisfy the defined
SAT-GRD spatial criterion (see Section 5) the best metrics are found for the red BRC residing closer
to the station site.

**7. Discussion on Cal/Val aspects and recommendations**

Throughout this assessment analysis, several critical points have been identified and
highlighted that should be addressed adequately towards a comprehensive Cal/Val study of the
Aeolus L2A aerosol products. These aspects, summarized in the current section, can: (i) serve as
guidelines for future relevant studies, (ii) improve our understanding about the advantages/limitations
of Aeolus data in terms of their usefulness and applicability in aerosol-related studies and (iii) suggest
possible upgrades regarding ALADIN's observational capabilities, the considerations of the applied
retrieval algorithms and the content of information in Aeolus L2A data.
A fair comparison of Aeolus L2A backscatter versus linear-derived retrievals acquired from
ground-based lidars, when depolarizing particles are recorded, requires the conversion of the latter
ones to circular co-polar (Aeolus-like) following Paschou et al. (2021). Nevertheless, it should be
acknowledged that the theoretical assumptions can be invalid either due to the orientation of the
suspended particles (e.g., mineral dust; Ulanowski et al., 2007; Daskalopoulou et al., 2021; Mallios
et al., 2021) or due to multiple scattering effects within optically thick aerosol layers (Wandinger et
al., 2010). The lack of discrimination between aerosols and clouds in Aeolus L2A data forces the
synergistic implementation of ancillary data in order to remove cloud contaminated Aeolus profiles
from the collocated sample with the cloud-free ground-based profiles. Nevertheless, it should be
noted that the cloud removal itself is not perfect. In our case, we are relying on MSG-SEVIRI cloud
observations, which are available at high temporal frequency (every 15 min) thus allowing a very
good temporal collocation with Aeolus. The indirect cloud-mask filtering applied to our analysis,
leads to a substantial improvement of the level of agreement between spaceborne and ground-based





retrievals. Despite its success, our proposed approach provides a sufficient and acceptable solution,
but undoubtedly cannot be superior to the utility of a descriptive classification scheme on Aeolus
retrieval algorithms similarly done in CALIOP-CALIPSO (Liu et al., 2019; Zeng et al., 2019).
Aeolus retrievals are available at coarse along-track resolution (~90 km). This imposes
limitations on their evaluation against point measurements, which are further exacerbated at sites
where the heterogeneity of aerosol loads in the surrounding area of the station is pronounced, taking
into account that the spatial collocation between spaceborne and ground-based retrievals is not exact.
Numerical outputs from reanalysis datasets (e.g., MERRA-2, CAMS) can be utilized as an indicator
of aerosols' burden horizontal variation, taking advantage of their complete spatial coverage, their
availability at high temporal frequency and their reliability (Innes et al., 2019; Gueymard and Yang,
2020). Over areas with a complex terrain, due to the coarse BRC horizontal resolution, they can be
recorded vertical inconsistencies between ground-based and satellite profiles (reported above ground
where its height is defined with respect to the WGS 84 ellipsoid), not physically explained. For the
derivation of the evaluation scores, it is required a rescaling of the ground-based profiles, acquired at
finer vertical resolution, in order to match the dynamically defined Aeolus' range bin settings.
Nevertheless, due to this transformation, the shape of the raw ground-based profile can be distorted
and the magnitude of the retrieved optical properties can be modified substantially thus affecting the
evaluation metrics. This artifact is evident in cases where the vertical structure of the aerosol layers
is highly variable thus hindering Aeolus capability to reproduce accurately their geometrical features
due to the coarse vertical resolution in which optical products are derived. Finally, the consideration
of backward trajectories can assist the characterization of the probed atmospheric scene by Aeolus.
Potentially, they can be also used as an additional criterion for the optimum selection of Aeolus BRC
for the collocation with the ground-based measurements. However, possible limitations may arise due
to temporal deviations among FLEXPART run, the Aeolus overpass and ground-based retrievals,
which might be critical taking into account the strong spatiotemporal variability of aerosol loads
across various scales.

**8. Conclusions**

The limited availability of vertically resolved aerosol products from space constitutes a major
deficiency of the Global Observing System (GOS). The launch of the Aeolus ESA satellite was a
major step towards this direction whereas the forthcoming EarthCARE satellite mission (Illingworth
et al., 2015) will accelerate further these efforts. ALADIN, the single payload of the Aeolus satellite,
constitutes the first UV HSRL Doppler lidar ever placed in space and it is optimized to acquire HLOS
wind profiles towards advancing numerical weather prediction (Rennie et al., 2021). ALADIN also





retrieves independently the extinction and backscatter coefficients of aerosols and clouds (grouped as
particulates according to Aeolus' nomenclature) via the implementation of the SCA algorithm.
The current work focuses on the assessment analysis of L2A particle backscatter coefficients
versus ground-based retrievals acquired routinely by lidar systems operating in Athens, Thessaloniki,
and Antikythera. The aforementioned stations contribute to the PANACEA Greek National Research
Infrastructure (Greek ACTRIS component) and to the European Aerosol Research Lidar Network
(EARLINET; Pappalardo et al., 2014). Overall, 46 cases are analyzed out of which 12 have been
identified in the urban site of Athens, 16 in Thessaloniki and 15 in the remote site of the Antikythera
island.
In the first part of the analysis, focus was given on the assessment of Aeolus L2A particle
backscatter coefficient, under specific aerosol scenarios, versus the corresponding measurements
obtained at the Antikythera island (southwest Greece). As expected, the misdetection of the cross
polarized lidar return signals, induces an underestimation of Aeolus L2A backscatter when
depolarizing mineral particles are probed. By converting the ground-based linear-derived total
backscatter to circular co-polar (i.e., Aeolus-like) it is computed that these underestimations range
from 13% to 33% (case of $10^{th}$ July 2019). For the case of $3^{rd}$ July 2019, when aerosol loads of
moderate intensity, consisting mainly of spherical particles, are confined below 4 km and they are
homogeneous in the surrounding area of the station, Aeolus is capable in reproducing quite well the
ground-based profile in terms of shape and magnitude. On the contrary, in the case of $8^{th}$ July 2020,
when the stratification of aerosol layers, detected up to 4 km by Polly$^{XT}$, becomes complex, Aeolus'
performance reveals an altitude dependency, probably attributed to the coarse vertical sampling of
the atmosphere. Finally, the agreement between Aeolus and Polly$^{XT}$ backscatter retrievals varies with
height on $5^{th}$ August 2020 when non-spherical particles (2-4 km) reside on top of a layer consisting
of spherical aerosols.
From our statistical assessment analysis, it has been revealed that the removal of cloud
contaminated spaceborne profiles, achieved via the synergy with MSG-SEVIRI cloud observations,
results in a significant improvement of the product performance. Unfortunately, the poor evaluation
metrics at the lowermost bins (attributed to either the surface reflectance or the increased noise levels
for the Aeolus retrievals and to the overlap issues for the ground-based profiles) are still evident after
the cloud filtering procedure. Between the two Aeolus vertical scales, the computed evaluation
metrics do not provide strong evidence of which of them performs better. Among the three stations
(ATH, ANT, THE) considered here, the best agreement was found in the remote site of Antikythera
island (spatially homogeneous AODs) in contrast to the urban sites of Athens and Thessaloniki. All


key cal/val aspects have been discussed thoroughly serving as guidelines and potential
recommendations for future studies.

The lack of the cross-polar channel downgrades ALADIN's performance under depolarizing

atmospheric scenes (e.g., dust, cirrus crystals, volcanic ash) hampering an effective aerosols/clouds
discrimination (Flamant et al., 2021). According to preliminary CAMS assimilation experiments
(A3S), relying on Aeolus L2A backscatter, it has been demonstrated to have a beneficial impact on
short-term forecasts. However, it is under investigation if the inclusion of the cross-polar channel will
expand these positive feedbacks on NWP (main scientific goal of the Aeolus satellite mission), taking
into account that aerosol-radiation interactions affect atmospheric dynamics and vice-versa. Another
important aspect is the coarse resolution of Aeolus L2A retrievals, both in horizontal and vertical,
imposing several limitations in an appropriate assessment analysis whereas it can be critical in their
implementation on other applications (e.g. data assimilation).

In the current study, we emphasized only on the particle backscatter coefficient due to the

limited number of ground-based extinction profiles. A wider assessment analysis is ongoing in the
framework of the Aeolus L2A Cal/Val study performed within EARLINET. Taking into account the
challenges that the SCA algorithm faces in retrieving the extinction coefficient reliably (Flament et
al., 2021), new retrievals will become available in the Aeolus L2A product. These include the
Maximum Likelihood Estimate (MLE) (Ehlers et al., 2021) and the EarthCARE derived AEOL-FF
and AEL-PRO products (refer to the latest L2A product release documentation). Finally, the best
assessment of Aeolus L2A products is expected versus the purpose-built eVe lidar (Paschou et al.,
2021) implemented in a dual-laser/dual-telescope configuration enabling the simultaneous emission
of linearly and circularly polarized radiation at 355 nm and the detection of the elastically
backscattered radiation with polarization sensitive channels as well as the inelastic (Raman)
backscattered radiation at 387 nm. As such, eVe can mimic Aeolus' observational geometry and test
the validity of the theoretical formulas applied for the derivation of the Aeolus-like backscatter from
the linearly polarized emission ground-based systems. The first correlative Aeolus-eVe
measurements have been performed in the framework of the Joint Aeolus Tropical Atlantic Campaign
(JATAC), that took place in Cape Verde in September 2021. Correlative measurements are also
planned in June 2022 at the same place during the ESA-ASKOS experimental campaign. The
geographical location of Cape Verde, situated on the "corridor" of the Saharan transatlantic transport
(Gkikas et al., 2022), is ideal for assessing Aeolus performance when non-spherical mineral particles
from the nearby deserts are advected westwards.




**Acknowledgments**

Antonis Gkikas was supported by the Hellenic Foundation for Research and Innovation (H.F.R.I.) under the "2nd Call for H.F.R.I. Research Projects to support Post-Doctoral Researchers" (project acronym: ATLANTAS, project number: 544). Vassilis Amiridis acknowledges support from the European Research Council (grant no. 725698; D-TECT). NOA members acknowledge support from the Stavros Niarchos Foundation (SNF). We acknowledge support of this work by the project "PANhellenic infrastructure for Atmospheric Composition and climatE change" (MIS 5021516) which is implemented under the Action "Reinforcement of the Research and Innovation Infrastructure", funded by the Operational Programme "Competitiveness, Entrepreneurship and Innovation" (NSRF 2014-2020) and co-financed by Greece and the European Union (European Regional Development Fund). We thank the ACTRIS-2 and ACTRIS preparatory phase projects that have received funding from the European Union's Horizon 2020 Framework Program for Research and Innovation (grant agreement no. 654109) and from European Union's Horizon 2020 Coordination and Support Action (grant agreement no. 739530), respectively. This research was also supported by data and services obtained from the PANhellenic Geophysical Observatory of Antikythera (PANGEA) of the National Observatory of Athens (NOA). We acknowledge support by ESA, in the framework of the Aeolus+Innovation (Aeolus+I) call, under Contract No. 4000133130/20/I-BG//.

**Data availability**

Aeolus Baseline 10 and 11 L2A data were obtained from the ESA Aeolus Online Dissemination System available at https://aeolus-ds.eo.esa.int/oads/access/.

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





**Table 1:** Statistical metrics for the unfiltered (clouds plus aerosols) Aeolus L2A SCA and SCA mid-bin backscatter
profiles at each PANACEA site.

| | SCA | | | | | SCA_mid_bin | | | | |
|---|---|---|---|---|---|---|---|---|---|---|
| **Station** | **Counts** | **Bias** | **Rel. Bias (%)** | **R** | **RMSE** | **Counts** | **Bias** | **Rel. Bias (%)** | **R** | **RMSE** |
| **ANT** | 255 | 0.06 | 13.63 | 0.49 | 1.14 | 173 | 0.25 | 45.59 | 0.57 | 1.01 |
| **ATH** | 60 | 0.73 | 199.65 | 0.49 | 2.26 | 43 | 1.16 | 272.84 | 0.52 | 3.10 |
| **THE** | 222 | 0.83 | 185.16 | 0.34 | 2.60 | 140 | 1.10 | 224.65 | 0.32 | 2.19 |


**Table 2:** As in Table 1 but for the filtered (only aerosols) Aeolus backscatter retrievals.

| | SCA | | | | | SCA_mid_bin | | | | |
|---|---|---|---|---|---|---|---|---|---|---|
| **Station** | **Counts** | **Bias** | **Rel. Bias (%)** | **R** | **RMSE** | **Counts** | **Bias** | **Rel. Bias (%)** | **R** | **RMSE** |
| **ANT** | 94 | -0.10 | -26.57 | 0.55 | 0.78 | 57 | 0.06 | 13.35 | 0.86 | 0.43 |
| **ATH** | 12 | 1.08 | 483.36 | 0.75 | 3.33 | 9 | 0.73 | 312.67 | 0.82 | 1.41 |
| **THE** | 133 | 0.46 | 130.49 | 0.39 | 1.86 | 81 | 0.55 | 145.08 | 0.43 | 1.20 |











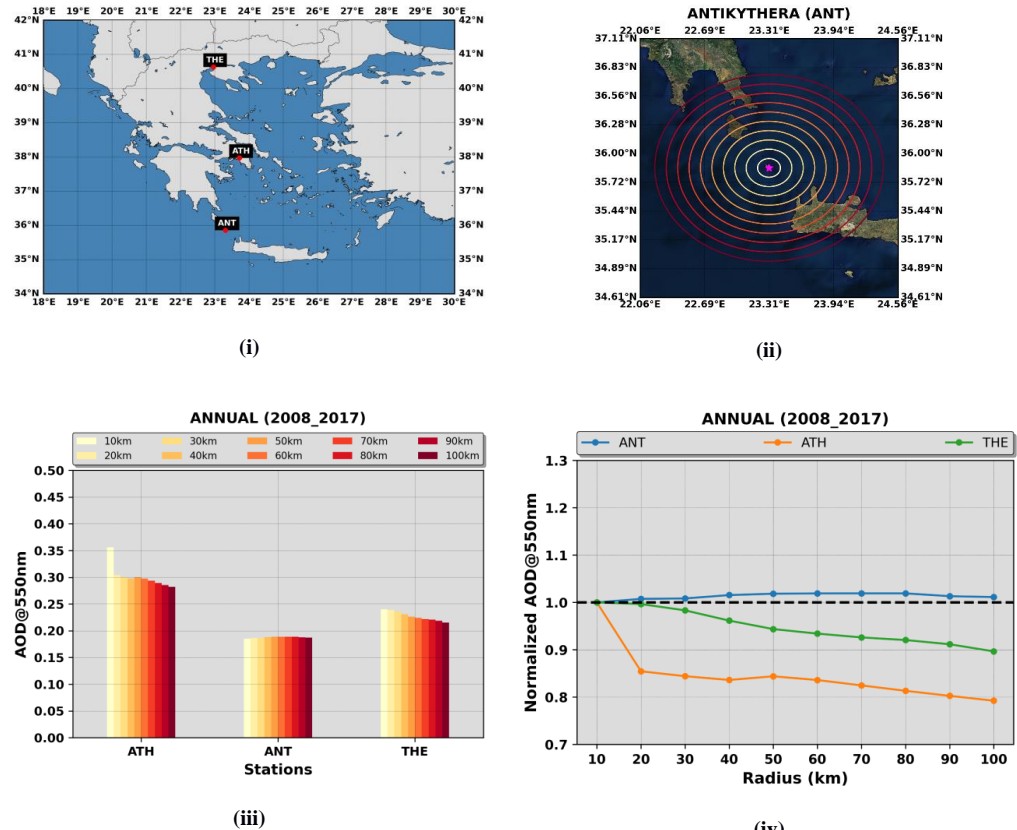

**Figure 1:** (i) Locations of the three Greek PANACEA sites, namely Athens (ATH), Antikythera (ANT) and Thessaloniki (THE), (ii) Concentric circles, around the Antikythera island, of radii from 10 to 100 km with an incremental step of 10 km, (iii) Climatological MODIS-Aqua AOD levels, representative for the period 2008 – 2017, for each circle area centered at each PANACEA site, (iv) Normalized climatological AODs for each circle area with respect to the corresponding levels of the inner circle.

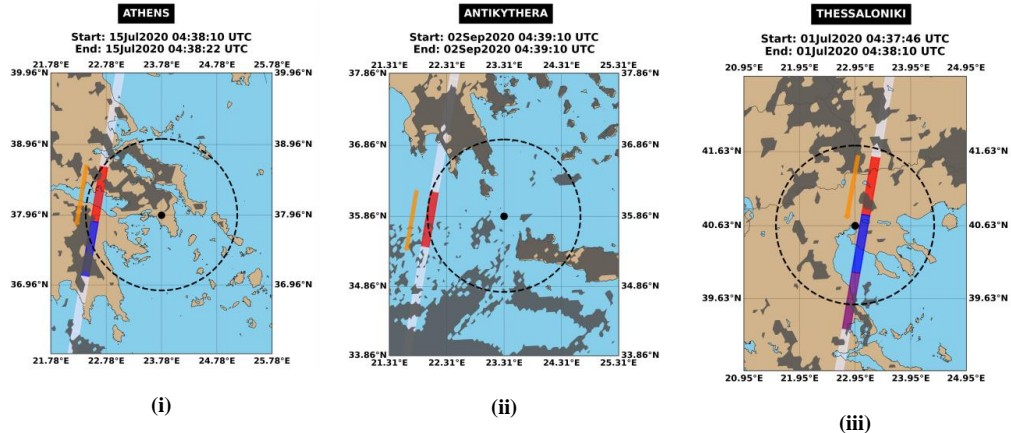

**(i)**        **(ii)**        **(iii)**

**Figure 2:** The white stripe indicates the ALADIN's measurements track and the colored rectangles correspond to the
Aeolus observations (~90 km along-track averaged measurements) falling within a radius of 120 km (dashed black line)
of the PANACEA stations (black dot). The orange arrow shows the Aeolus flight directions (ascending or descending
orbit). Dark grey shaded areas: MSG-SEVIRI cloud mask product (CLM) at the nearest time to Aeolus overpass. The
start and end time (in UTC) of the ALADIN observations are given in the title of each plot.

**Figure 3:** Vertical profiles of backscatter coefficient at 355 nm acquired by ALADIN for the Level 2A SCA (regular vertical observation grid, brown solid curve) and SCA mid-bin (reduced vertical observation grid, black solid curve) products. The dashed lines correspond to the estimated SCA backscatter coefficient errors (brown) and SCA mid-bin backscatter coefficient errors (black). Vertical profile of Polly[XT] backscatter coefficient (pink solid curve) at UV wavelength (355 nm) and associated errors (pink shaded area). Polly[XT] Aeolus-like backscatter coefficient (light-blue





solid curve) after converting the linear-derived products to circular co-polar according to Paschou et al. (2021). The
ground-based profiles have been acquired at the Antikythera station (southwest Greece) on: (i) 10th July 2019, (ii) 3rd July
2019, (iii) 8th July 2020 and (iv) 5th August 2020. The red color font denotes which Aeolus BRC (along with the overpass
time) has been selected based on the defined collocation criteria.

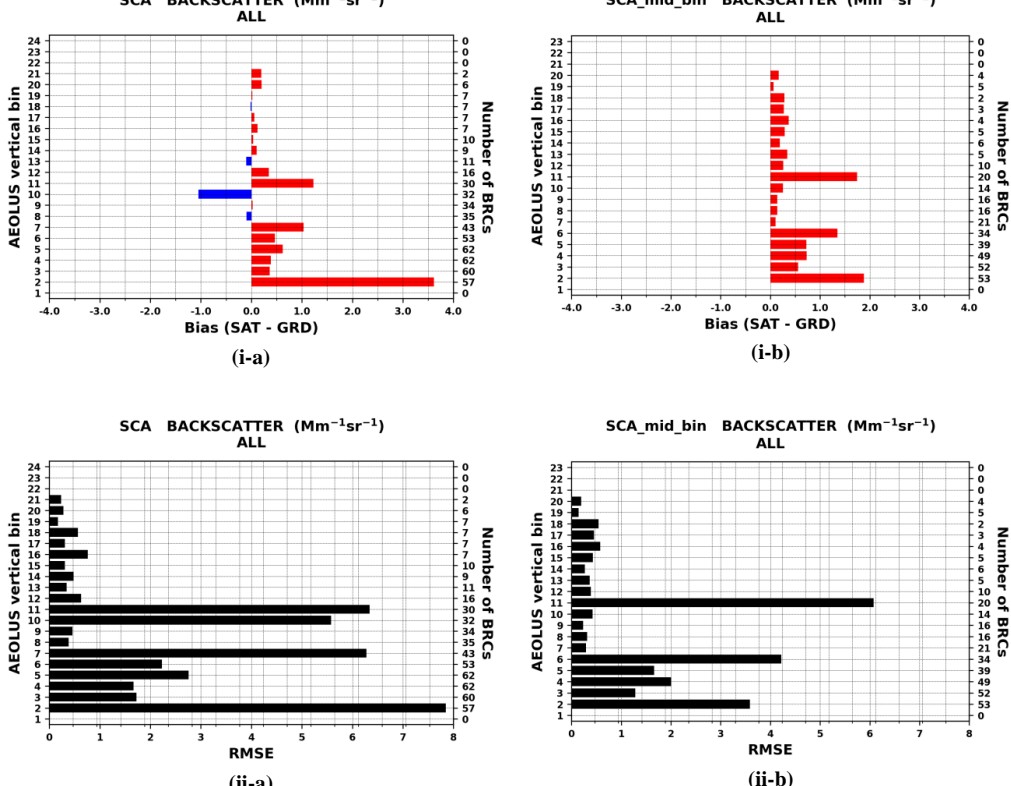

**Figure 4:** Bias (i) and root mean square error (ii) metrics for the unfiltered Aeolus L2A backscatter retrievals reported at
the regular (a) and mid-bin (b) vertical scales. The biases are defined as SAT-GRD and the positive/negative departures
are depicted with red/blue bars. The statistical metrics are vertically resolved based on Aeolus bins indices (left y-axis).
The number of BRCs participating in the metrics calculations at each bin are given on the right y-axis.











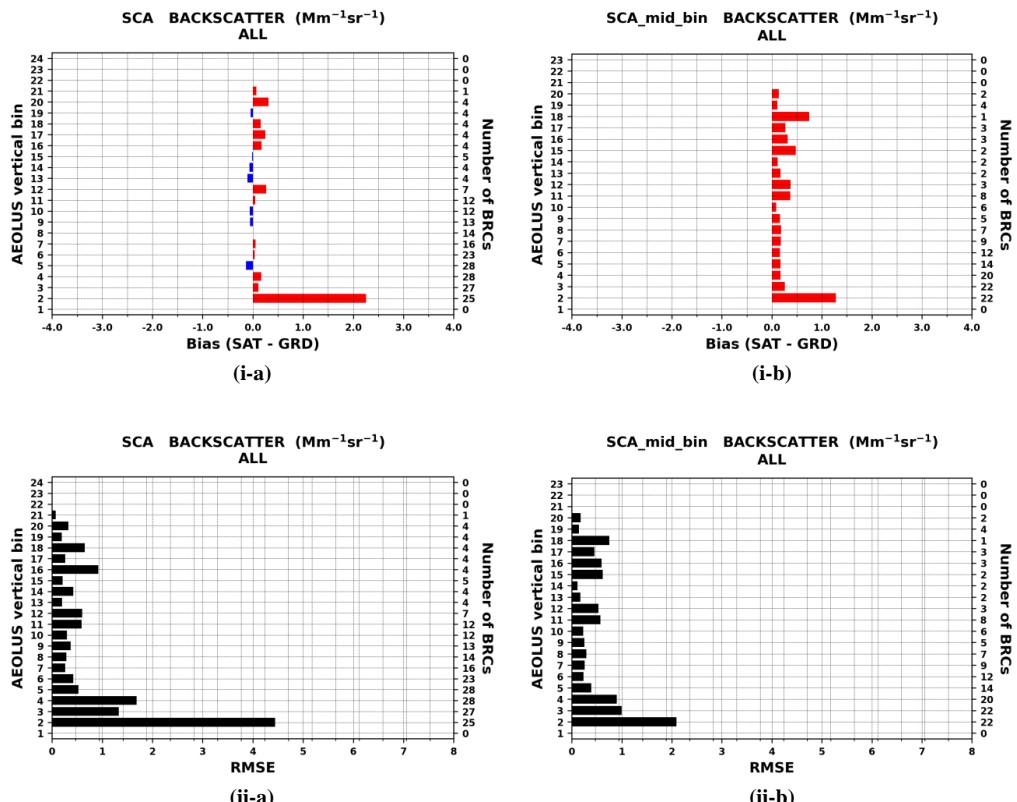

**Figure 5:** As in Figure 4 but for the filtered Aeolus L2A backscatter retrievals.

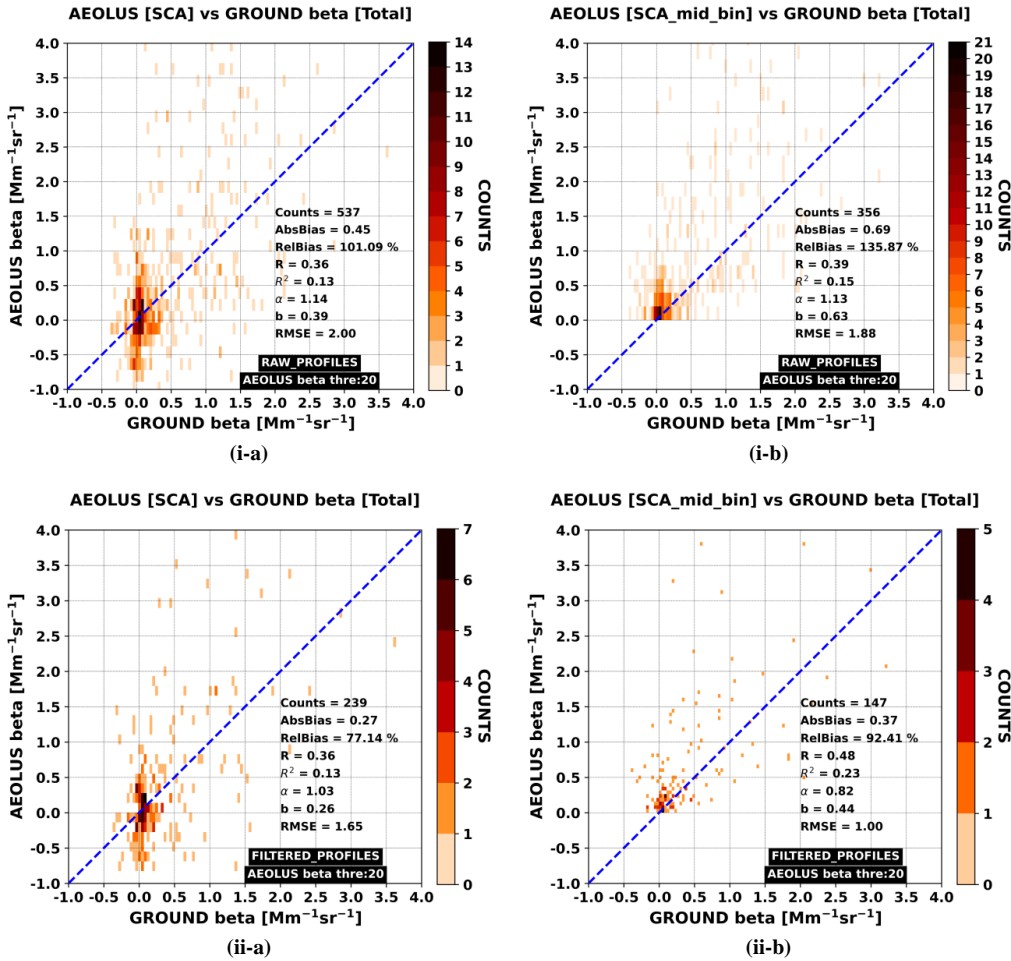

**Figure 6:** 2D histograms between Aeolus (y-axis) and ground-based (x-axis) backscatter coefficient retrievals. In the upper (i) and bottom (ii) panels are depicted the results for the cloud+aerosol backscatter (unfiltered) and cloud-cleared backscatter (filtered) Aeolus profiles, respectively. On the left and right columns are illustrated the results corresponding to Aeolus regular (24 bins) and mid-bin (23 bins) vertical scales, respectively. Aeolus backscatter values larger than 20 Mm$^{-1}$ sr$^{-1}$ are masked out from the collocated sample.



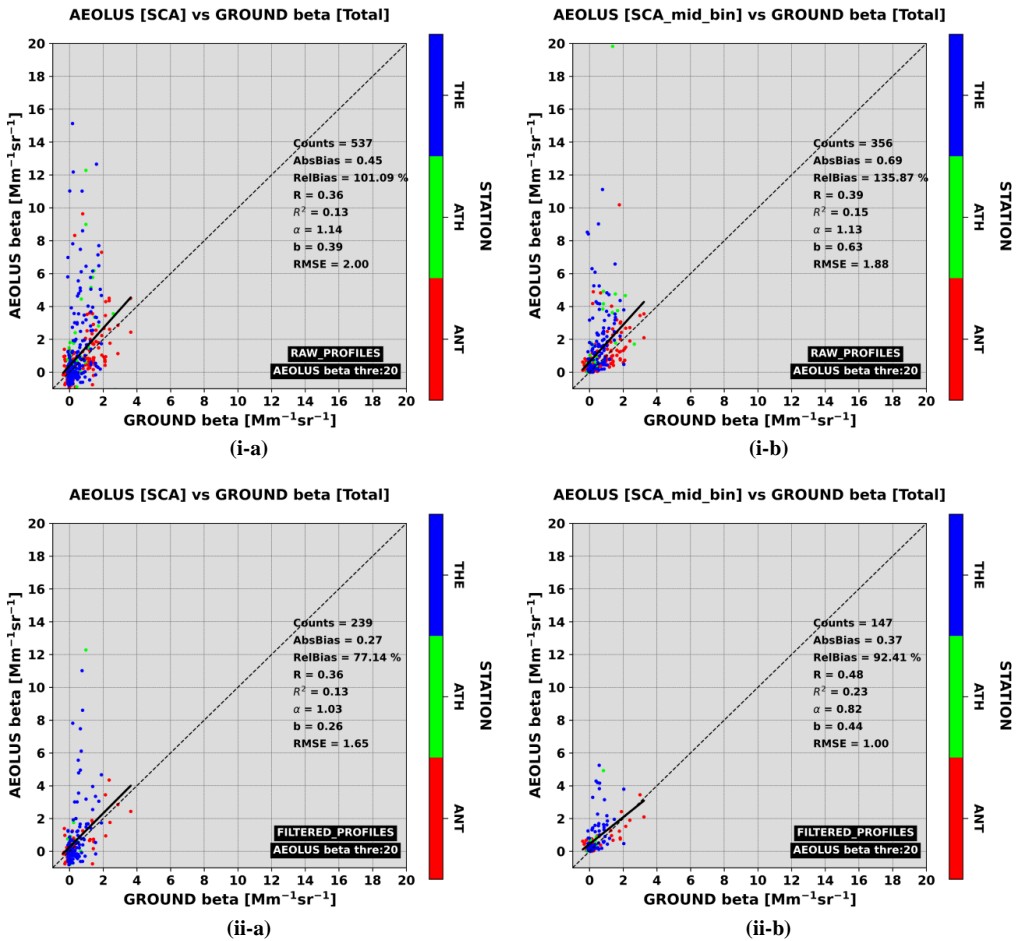

**Figure 7:** Scatterplots between Aeolus (y-axis) and ground-based (x-axis) backscatter coefficient retrievals resolved based on the indices of Aeolus vertical bins (colored circles). In the upper (i) and bottom (ii) panels are depicted the results for the unfiltered and filtered Aeolus profiles, respectively. On the left and right columns are illustrated the results corresponding to Aeolus regular (24 bins) and mid-bin (23 bins) vertical scales, respectively. Aeolus backscatter values larger than 20 Mm$^{-1}$ sr$^{-1}$ are masked out from the collocated sample.