# Peer review of "First assessment of Aeolus SCA particle backscatter coefficient retrievals in the Eastern Mediterranean"

_Atmospheric Measurement Techniques, 2022_

## Author Response (AR1)

We would like to thank the Reviewer for his/her thorough and detailed review as well as for the suggested papers. Our replies (regular font) for each comment (bold font) are provided below.

**Reviewer #1**

**The authors assess the particle backscatter coefficient profiles from Aeolus/ ALADIN using co-located ground-based lidars at three locations in Greece, together with auxiliary model and satellite datasets. They attempt to attribute discrepancies between space borne and ground based lidars to (i) the natural variability of aerosols, (ii) instrument or retrieval limitations and/or (iii) spatial temporal co-location issues. This paper needs substantial improvements before it is worthy of publication in AMT.**

**Major comments**

**In a very general sense, numerous sentences throughout the paper are too wordy and their structure just too complicated. For example, "are utilized towards an optimum characterization of the probed atmospheric conditions under the absence of a classification scheme in Aeolus profiles" could be replaced by something along the lines of "are used to characterize the atmosphere as Aeolus/ALADIN does not provide an atmospheric classification product". This makes for a lengthy paper. Other examples are "obtained results" instead of simply writing "results" or "probed Atmosphere" instead of "atmosphere" throughout the paper.**

We would like to thank the reviewer for his/her suggestions. In the revised manuscript, we have tried to reduce lengthy sentences and to "simplify" them thus improving the readability of the manuscript.

**We recommend that the authors:**

**Use ALADIN in all caps or Aeolus/ALADIN consistently (instead of small caps and/or interchangeably using both names). Also, they should spell it out once when first introduced (i.e., "Atmospheric LAser Doppler Instrument").**

Done.

**Add a table describing the lidar(s) at each of the three locations. This table could contain, for example, the lidar's name, a small description, its limitations and uncertainties, its products. It could also contain information on how cloud screening is performed and possibly a dominant type of aerosols at each location (together with references related to these stations).**

We believe that it is already provided the information requested by the reviewer. We would like to remind that in the submitted manuscript, there is a dedicated section discussing all the necessary technical aspects of each lidar system, the possible instrument deficiencies, the SCC automatic processing chain for lidar data (according to EARLINET) as well as the inversion methods applied during daytime and nighttime.

**Add a brief description of the two models used in this study and, especially, their limitations. The way the analysis is written might sometimes give the impression that model results are considered as accurate as observations.**

We would like to avoid extending the length of the manuscript by adding information for the CAMS and MERRA-2 since are the two most well-known reanalysis datasets widely used in aerosol (and atmospheric in general) studies. Nevertheless, we agree with the reviewer that our description in the submitted manuscript was not pretty clear and maybe there was a confusion regarding the way the reanalysis products are used in our study. In the revised text, we are clarifying that the aforementioned data are used just as an indicator of the aerosol load in the surrounding area of the station synergistically with the AERONET and Polly$^{XT}$ observations, which always serve as the "truth". Below we are providing the relevant part in Section 7 in the revised manuscript.

*"Numerical outputs from reanalysis datasets (e.g., MERRA-2, CAMS) can be utilized as an indicator of aerosols' burden horizontal variation, taking advantage of their complete spatial coverage, their availability at high temporal frequency and their reliability in terms of total AOD (Innes et al., 2019; Gueymard and Yang, 2020). Nevertheless, such data are better to be utilized in a qualitative rather, particularly in terms of aerosol species, than a quantitative way since they cannot be superior of actual aerosol observations."*

**Add more text comparing their results to those in studies such as Baars et al. (2021) and Abril-Gago et al. (2022). In a more general sense, it would be helpful to open the paper by describing in more detail what their study brings to the table/how it complements other studies.**

We have modified this part of the manuscript as suggested by the reviewer. Below is given the revised text

*"Abril-Gago et al. (2022) performed a statistical validation versus ground-based observations from three Iberian ACTRIS/EARLINET lidar stations affected mainly by dust and continental/anthropogenic aerosols. In their Cal/Val study, they processed AERONET optical properties related to particles' size and nature along with HYSPLIT air-mass backtrajectories towards characterizing the prevailing aerosol conditions."*

**Specify the way they average the MODIS AOD and add more analysis to their spatial characterization of aerosols at the three different stations. We believe that there is more to the characterization of aerosol spatial variation than simply averaging the AOD in different boxes. The authors should refer to previous studies such as Anderson et al., (2003), Sayer and Knobelpiesse (2019) or Shinozuka and Redemann (2011).**

In the revised manuscript, we have clarified better the MODIS AOD averaging and have extended the analysis discussing the coefficient of variation and the spatial autocorrelation.

**Discuss the limitations of their cloud screening of the Aeolus/ALADIN aerosol backscatter profiles using SEVIRI.**

According to the [product user guide](), the quality of the CLM (Cloud Mask) is impacted by the following limitations:

1. The ECMWF temperature and humidity fields are not interpolated in time and space. This means that for all pixels within each segment the same temperature and humidity profile for the period for which the forecast has same validity time. This may lead to artificial straight lines in the display of the product
2. Within the MTP MPEF (Meteorological Product Extraction Facility) algorithms, a 10 level ECMWF temperature profile is used for the determination of e.g. the atmospheric correction, impacting the cloud detection especially in the lower troposphere

We have added a short description in the revised manuscript regarding the limitations of the SEVIRI CLM product. Please note that we are not mentioning the Meteosat-7 calibration bias with respect to MSG or IASI calibrations since these data are not relevant for our study period.

**Discuss whether the Aeolus/ALADIN, with its 87 km horizontal resolution, is in fact able to characterize the aerosol natural variability at the three locations.**

For the AOD spatial variability in the surrounding area of Antikythera, we are using the outputs from the MERRA-2 and CAMS reanalyses. Both products are derived at coarse spatial resolution (please see relevant plots in the supplementary material). This makes quite difficult to investigate the variability within the Aeolus BRC. An alternative approach would be to exploit MODIS L2 AODs, which are derived at fine spatial resolution

(10 km x 10 km). However, MODIS-Aqua crosses the region of interest once per day around 11-12 UTC and there is a non-negligible temporal departure with the Aeolus overpass time (both for dawn and dusk orbits). Therefore, since we don't have sufficient information (data available at fine spatial and temporal resolution would be suitable) we are not able to address adequately the comment raised by the reviewer.

**Avoid strong statements such as "very good performance" when it pertains to a specific altitude, one case study and no quantification of the differences between space borne and ground based lidar in that case.**

We have modified accordingly the text.

**Use additional satellite derived aerosol information (e.g., CALIOP, TropOMI) to further characterize the aerosol during their case studies.**

We haven't found coincident Aeolus-CALIPSO overpasses for the selected study cases. It is in our plans to incorporate in the future Aerosol Index and Aerosol Layer Height observations from TROPOMI.

**Specify why they use AERONET level 1.5 instead of more accurate level 2 data (quality assured). Also, they should show AERONET-derived FMF and SSA at different wavelengths throughout the day (in addition to what they already do -- spectral AOD and angstrom exponent). This would be like Figure 8 in Abril-Gago et al. (2022). Let us remind the authors that, in addition to a size difference, a difference in SSA at two wavelengths from AERONET could point to the presence of dust versus smoke (e.g., Russel et al., 2014; Kacenelenbogen et al., 2022).**

We would like to thank the reviewer for his/her comment. In our analysis we are using AERONET Level 2.0 data but there was a typo both in the manuscript as well as in the relevant plots in the supplement. This has been corrected in the revised manuscript and in the supplement. Regarding the SSA we are aware for its spectral dependence under dust/smoke conditions as it is shown in the studies mentioned by the reviewer as well as in Gilles et al. (2012). However, this was not feasible in our analysis due to the very limited availability of SSA retrievals both for Level 1.5 and 2.0 as it shown in plots below.

[Figure]

[Figure]

Therefore, in the revised supplement we are including only the FMF diurnal variation for each case and we have added the text accordingly.

**Specify why they show Aeolus/ALADIN profiles that are cloud contaminated (or "unfiltered") in their analysis. It is not obvious why there would not be more disagreement between cloud-unfiltered space-borne and cloud-filtered ground-based profiles compared to cloud-filtered space and ground profiles. Or is this a way to test their SEVIRI-based cloud filtering method?**

Under the presence of clouds, the backscatter coefficient is significantly higher than those when aerosols are recorded. Moreover, there are differences between the cloud and aerosol layers' structure. Based on these facts, we are expecting and we see differences between the cloud unfiltered and filtered profiles. It is well-known that in the raw L2A profiles there is not a discrimination between aerosols and clouds. Therefore, presenting the results from the comparison of the unfiltered Aeolus profiles against those acquired by the ground-based lidars, we are highlighting that this deficiency is critical and can lead in erroneous conclusions. This is further stretched when we are contrasting the results shown in Figures 4 (unfiltered) and 5 (filtered). At a first level, our intention is to highlight the necessity of including ancillary cloud information (SEVIRI in our case) for an appropriate assessment of Aeolus profiles. However, the most important is to highlight the imperative need to deploy a cross-channel, in a possible Aeolus follow on mission, that will facilitate a feature classification scheme, similarly done in the CALIPSO observations. We believe, that the above-mentioned aspects are well stated in our manuscript.

**Figures should be called in the order they appear.**

Done.

**Shorten the conclusion.**

We have reduced the conclusions.

**Detailed comments:**

**Title: The authors might want to add "ALADIN" and "aerosol" in the title for increased searchability**

We prefer to keep the original title

**Line 29: Why not give examples after "a variety of aerosol species."**

We think that it is better to focus on the main aspects of the study. Moreover, in the introduction there is paragraph discussing all the aerosol types encountered over the broader Greek area.

**Line 32: Why is PANACEA spelled out but not AERONET, CAMS, MERRA-2 etc...? We recommend either spelling none or all of them.**

Because PANACEA, which is the Greek National RI, is not well-known to the community in contrast to AERONET, CAMS and MERRA-2.

**Line 33: we recommend writing "sunphotometry observations "... "model reanalysis" ..." modeled air mass back trajectories"**

We think that it is already well stated in the submitted manuscript.

**Line 36: Again, multiple sentences throughout the paper are too wordy. For example, "are utilized towards an optimum characterization of the probed atmospheric conditions under the absence of a classification scheme in Aeolus profiles" could be replaced by something along the lines of "are used to characterize the atmosphere as Aeolus/ALADIN does not provide an atmospheric classification product".**

We believe that the sentence is well written and its meaning is pretty clear.

**Line 40: "very good" is too strong a statement here.**

We have replaced with "good".

**Line 44: We recommend writing "46 identified cases when using [this time frame] at all three stations...".**

We prefer to keep our initial version.

**Line 47: "positive tendency" could be replaced by "improvement" and "both Aeolus vertical scales" by "multiple Aeolus vertical scales"**

We agree with the first part of the reviewer's comment. The word "multiple" is misleading since there are only two Aeolus vertical scales.

**Line 48: we recommend to replace "justified" by "explained" + "in the vertical the Aeolus performance"**

We believe that the use of the word "justified" is appropriate and fits well in the sentence.

**Line 49: "performance decreases" followed by the explanation for that decrease is not clear.**

We have modified the sentence as follows:

*"In vertical, Aeolus performance downgrades at the lowermost bins due to either the contamination from surface signals or the increased noise levels for the Aeolus retrievals and to the overlap issues for the ground-based profiles."*

**Line 83: We recommend "Such observations are provided by networks... or by dedicated experimental airborne (Ansmann et al., 2011; Weinzierl et al., 2016) or shipborne campaigns (Bohlmann et al., 2018)".**

We would like to thank the reviewer for his/her suggestion but we prefer to keep the sentence in its current form.

**Line 87: We recommend "characterization of aerosol vertical structure at global (e.g., Liu et al., 2008) … was performed using CALIOP … and CATS… respectively on the CALIPSO and the ISS…". CATS could use other references such as Lee et al., (2019).**

We have added the reference as suggested by the reviewer! Thanks a lot!

**Line 108: "good quality" needs a reference.**

Done

**Line 116: ALADIN needs to be in all caps throughout.**

Done.

**Line 117: We find the description in Flament et al., (2021), a little clearer (e.g., "The UV laser beam is linearly polarized at the laser output. It goes through a quarter-wave plate before being routed towards the telescope and is thus transmitted towards the atmosphere with a circular polarization…").**

We have rephrased accordingly.

**Line 126-128: This is important and should be explained in more detail. This paper is about validating Aeolus/ ALADIN. The limitations of the lidar should be clearly explained and other papers should be referenced.**

There is a thorough discussion throughout the text regarding the limitations of the lidar whereas all the related papers are cited.

**Line 130: We recommend "ALADIN" and why are "continuous" calibration and validation needed? Please explain.**

We have changed "Aladin" to "ALADIN" and we have removed the word continuous.

**Line 136: We recommend "L2A aerosol optical properties".**

We do not agree with the reviewer's comment. Aeolus L2A optical properties refer both to aerosols and clouds (grouped as particulates) since there is not a separation between them.

**Line 138: Regarding the "excellent agreement" here, we recommend adding some nuance. These results for a case study with a strong non-depolarizing aerosol, were ~satisfying only between ~4 and 8km.**

We are clarifying better this sentence in the revised manuscript.

**Line 142 – 156: If the type of aerosols over the three regions is discussed here, then you might consider not repeating it elsewhere (e.g., section 4). In general, we recommend adding a table describing the lidar(s) at each of the three locations. This table could contain, for example, the lidar's name, a small description, its limitations and uncertainties, its products. It could also contain information on how cloud screening is performed from these ground-based lidars and possibly the dominant type of aerosols at each location (together with references related to these stations).**

In this paragraph we are providing a summary of the aerosol types encountered within the broader Greek area. Actually, our intension is to note the coexistence of various aerosol species thus highlighting that the region of interest is ideal for the purposes of the study. On the contrary, in Section 4 we are specifying the aerosol type(s), per case, by exploiting the ancillary datasets (i.e., models and observations). Therefore, we are discussing about two different things and there is not overlap between these two parts of the manuscript. Regarding the second part of this comment please see our reply above.

**Line 184: HSRL was already introduced on line 116.**

Thanks!

**Line 187: We recommend "are backscattered".**

We agree. Thanks!

**Line 206-215: What is the purpose of describing algorithms that are not used in the study (e.g., ICA and MCA)?**

Actually, it is not a description but a short mention of the ICA and MCA algorithms just to inform a reader, who is not so much familiar with Aeolus, for their existence. However, we have removed this part in the revised manuscript.

**Line 216: We recommend "the primary and most reliable".**

We have modified this part in the revised text.

**Line 217: We recommend "measured signals in the Mie channel".**

Done.

**Line 224: We recommend "signals in each channel"; also, the sentence is not clear.**

This paragraph explains the cross-talk correction which is required due to the "cross-contamination" between the Rayleigh and Mie channels. The first sentence serves as a short statement whereas the following sentences describe explicitly the cross-talk issue attributed to the ALADIN HSRL instrument design.

**Line 260 and section 4: Again, the three stations, type of lidar(s), products, uncertainties, limitations (e.g., overlap), etc. could really use a table. That table could also show a predominant aerosol type over the region and a median and standard deviation AOD from satellite(s).**

We think that we have already provided a sufficient answer to this comment raised by the reviewer. Please see our previous replies.

**Line 263: We recommend using "different" instead of "adverse".**

Done.

**Figure 1-i: It would be helpful to write "all three stations are within Xkm of each other".**

We are providing only here the distances between the stations, as requested by the reviewer, and not in the caption of Figure 1 since we don't see why such information is helpful. The distances between ANT-ATH, ANT-THE and ATH-THE are equal to 273 km, 531 km and 305 km, respectively.

**Line 272 to 275: The authors must mean "to ensure the consistency of all lidar-derived observations"?**

Exactly! We have slightly rephrased this sentence.

**Line 279: We recommend deleting "measurements" here.**

We think that the word "measurements" fits well in the sentence.

**Line 280-281: Why is this assumption plausible? Does it remain to be tested?**

A well-mixed planetary boundary layer (PBL) is a common assumption, especially during afternoon hours above land when as mentioned in Stull, (1988; page 450, 11.2.1 The Mixed Layer Profile Shapes): *"…intense vertical mixing tends to leave conserved variables such as potential temperature and humidity nearly constant with height (see Fig 11.1). Even wind speed and direction are nearly constant over the bulk of the mixed layer."* Hence, the concentration of pollutants (similar to humidity) can be assumed to be nearly constant with height.

Under this assumption and in certain cases, lidar profiles can be linearly extended to the ground below the full overlap region when no information is available (i.e. in cases when we need to compare aerosol optical depth (AOD) from a sun-photometer, to the lidar derived AOD). Nevertheless, the altitudes below the full-overlap region (~800m for the ground-based lidar systems used in our study) were *not* accounted for herein, since i) we did not want to introduce any bias in our results caused by linearly interpolating ground-based profiles to the ground, ii) the Aeolus performance downgrades near the surface either due to surface reflectance or increased noise levels at the lowermost bins.

Since these altitudes were not accounted for, we removed this sentence from the manuscript, as it might be confusing for the reader.

**Reference:** *Stull, R.B.* An Introduction to Boundary Layer Meteorology*; Springer: Berlin/Heidelberg, Germany, 1988*.

**Line 286-288: Doesn't this apply to all three stations? Also why not add biomass burning aerosols here?**

It was not well stated in the submitted manuscript and we have remove it in the revised text.

**Line 338: We recommend "Aerosol spatial variability in the vicinity of the PANACEA sites". A description of the dominant aerosol type at each station would fit well here but then the authors would have to delete it from the introduction to avoid repetition.**

We would like to thank the reviewer for his/her suggestion but we prefer to keep the initial title.

**Section 4.4: The purpose behind studying the spatial variability could be explained more clearly. Our understanding is that the authors are attempting to characterize spatial variability to explain a potential disagreement between Aeolus/ ALADIN and ground-based lidars. A disagreement could be due to imperfect spatial co-location and/or simply ALADIN's 87 km horizontal resolution. The authors are studying horizontal variability by using total column integrated AOD and that should be mentioned as well. There could be minimal horizontal variability but a strong vertical variability. It is also not clear how the authors have computed the mean AOD from MODIS. Is it a arithmetic or geometric mean? It does make a difference -- see e.g., Sayer and Knobelspiesse, (2019). Spatial characterization analysis usually uses mean and standard deviations within each satellite grid cell and/or the variation between consecutive satellite pixels or airborne measurements within a region (e.g., Anderson et al., 2003 or Shinozuka and Redemann et al, 2011).**

We think that it is pretty clear the purpose of this short analysis and it is already explicitly stated in the submitted manuscript. We are copying below the relevant part from the text.

*"The aim of this introductory analysis is to investigate the horizontal homogeneity of the aerosol optical depth (AOD) in the respective broader areas, playing a key role in the comparison of ground-based and spaceborne profiles, which are not spatially coincident as it will be shown in a following section (i.e., collocation method)"*

As correctly mentioned by the reviewer, the disagreement between spaceborne and ground-based profiles can be due to the imperfect spatial collocation and/or the coarse spatial resolution of Aeolus BRC. However, we would like to remind that both aspects are already discussed in the submitted manuscript. For instance, we are providing below a part from the submitted text related to the reviewer's comment.

*"Aeolus retrievals are available at coarse along-track resolution (~90 km). This imposes limitations on their evaluation against point measurements, which are further exacerbated at sites where the heterogeneity of aerosol loads in the surrounding area of the station is pronounced, taking into account that the spatial collocation between spaceborne and ground-based retrievals is not exact."*

In the revised text we are clarifying that the MODIS AODs correspond to the entire atmospheric column as well as that we are calculating the arithmetic mean. Regarding the vertical variability this is exactly what is examined in the current study either for specific cases or for the entire collocated sample. Please note that in the revised manuscript we have extended our analysis including the coefficient of variation and the spatial autocorrelation.

**Fig. 2: The orange arrow is hard to see; It is also not clear if the analysis involving the 46 cases considers the closeness of the actual track to the station (e.g., better spatial colocation on July 1st).**

We have changed the color of the arrow denoting Aeolus' flight direction. For each case, we are taking into account all BRCs residing within a circle area (of 120 km radius) centered at the station coordinates. Each case corresponds to a specific day and for this day the number of BRCs can be one, two or three (maximum). In the revised manuscript, we are clarifying better this point.

**Line 381: The authors should discuss this "temporal window extension" in more detail and attempt to explain its consequences.**

We have revised this part of the manuscript providing more details. Below is given the modified text.

*"For the ground-based observations, the aerosol backscatter profiles are derived considering a time window of ± 1 hour around the satellite overpass. Nevertheless, this temporal collocation criterion has been relaxed or shifted in few cases to improve the quality of the ground-based retrievals (i.e., by increasing the signal-to-noise ratio) as well as to increase the matched pairs with Aeolus L2A profiles. Both compromises are applied since the weather conditions favoring the development of persistent clouds may eliminate the number of simultaneous cases. It is noted, however, when the temporal window is shifted or relaxed we are taking into account the homogeneity of the atmospheric scene (probed by the ground lidar). For the Antikythera station we did not deviate from the pre-defined temporal criterion apart from one case study. In Thessaloniki and Athens, the time departure between Aeolus and ground-based profiles can vary from 1.5 to 2.5 hours. Overall, 43 cases are analyzed out of which 15 have been identified over Antikythera, 12 in Athens and the remaining 16 in Thessaloniki."*

**Line 385-397: We recommend "derived from radiances measured by SEVIRI", "indication of cloud presence"; the limitations of using this SEVIRI cloud mask should be discussed. For example, could SEVIRI be missing small broken water clouds? What about cirrus clouds? and what would be the consequences on the Aeolus/ ALADIN aerosol profiles?**

We have modified the "*derived from radiances measured by SEVIRI*" part of the sentence in the manuscript, as suggested by the reviewer. Regarding the limitations of the SEVIRI CLM product, please see our reply in a relevant comment above.

**Line 403: Regarding (ii), how will the authors differentiate the effects of natural variability, the imperfect co-location and the errors in the Aeolus/ ALADIN instrument? See e.g., section "nature versus noise" in Anderson et al., 2003. Regarding (iii), this was already demonstrated in numerous studies.**

Please read our previous replies in relevant comments. We are not claiming that point III is a novelty of our study.

**Line 406: We recommend "(...) Cal/Val study to facilitate the interpretation of our findings and to identify possible upgrades in the Aeolus/ALADIN retrievals."**

We agree with the reviewer's suggestion and we have modified the text accordingly.

**Line 409: We recommend "the results are depicted in Figure 3".**

We think that we do not have to change something here.

**Line 411: We recommend "... Aeolus retrievals are provided at a coarse horizontal and vertical resolution ..."**

Done.

**Line 420: We recommend "To depict the spatial patterns (...)".**

Done.

**Line 423: The fact that MERRA-2 and CAMS provide "aerosol products of high quality" is a strong statement and should be explained. The explanation should include model evaluation results from previous studies. Model aerosol optical properties and model aerosol speciation have serious limitations, which should absolutely be mentioned in the text.**

The two reference studies in the submitted manuscript show that the AOD from both reanalysis datasets is a product of high quality. For example, please see the CAMS evaluation metrics given in Tables 3.2.1 (North Africa, Middle East and Europe) and 3.3.1 (Mediterranean) in Errera et al. (2021). In Gueymard and Yang (2020), the AOD evaluation metrics at station level, both for MERRA-2 and CAMS, depicted in Figure 6 reveal a very good agreement with observations. We fully agree with the reviewer that there are limitations on the modelled aerosol optical properties and aerosol speciation. Nevertheless, we are not claiming that the aerosol speciation is accurate and we are clarifying that the aerosol outputs from CAMS and MERRA-2 reanalyses are used just as an indicator (this is explicitly stated in the manuscript) to support a better characterization of the probed atmospheric scene. Even though the evaluation of CAMS and MERRA-2 is beyond the scope of this paper, we would like to mention that for the selected cases (Section 6.1) it seems that their performance is quite good in terms of capturing the aerosol load and the presence of coarse/fine particles.

**Line 425: Why not use AERONET Level 2 (quality assured) instead of Level 1.5?**

Please see our reply in a relevant comment above.

**Line 427: We recommend "characterization of the aerosol load and size over the station"**

Done.

**Line 432: Figures should be called in the order they appear. Figure S4 is introduced before S1, S2 or S3.**

Done.

**Line 435: We recommend adding "at 550 nm".**

Done.

**Line 437: The truth should be in the sunphotometer direct measurements. This sentence, as written, could be interpreted as things being the over way around.**

We think that it is pretty clear the meaning of this sentence. We are saying that the MERRA-2/CAMS findings are confirmed by the AERONET observations and the PollyXT retrievals (the following sentence).

**Line 438: The Angstrom exponent should be briefly explained here (i.e., difference of AOD at two (or more?) wavelengths that informs on the particle size) and references for typical dust angstrom exponent should be added to the text (e.g., Dubovik et al., 2002).**

We have added the wavelength pairs (440-870nm) as well as the reference suggested by the reviewer.

**Figure S2: This10 min-worth of high VLDR content looks suspiciously high compared to the consecutive profiles in the curtain plot. How do the authors explain that dust was present for only 10min and then suddenly disappears?**

We thank the reviewer for this comment that we believe is referring to the area indicated by the red box in the figure. As explained in the legend of Fig. S2, the 10-minute interval that seems to present higher VLDR values, corresponds to the calibration of depolarization measurements automatically performed by all Polly$^{XT}$ lidar systems (Engelmann et al., 2016). The procedure followed is the ±45∘ -calibration method described in Freudenthaler et al. (2009) and Freudenthaler, (2016). As described in Engelmann et al. (2016), a polarizer placed in front of the detectors of the Polly$^{XT}$ lidar system, rotates automatically three times per day, first at -45 (for 5 minutes) and then at +45 (for 5 minutes) degrees with respect to the polarization plane of the laser beam. When operating in normal mode, measurements are performed without the polarizer into the light path. Furthermore, in order for the aerosol optical properties to be retrieved, this 10-minute time interval is excluded from the data.

[Figure]

**Figure S2:** Time-height plot of the Volume Linear Depolarization Ratio (VLDR) at 355 nm at PANGEA station during 10 July 2019, 18:00 - 23:59 UTC. Station elevation is at 193 m a.s.l. The time period between 21:30 and 21:40 UTC corresponds to routine depolarization calibration measurements of the Polly$^{XT}$ lidar system and is indicated by a thick black vertical line on the plot. Below 6 km, VLDR values (5 - 10%) indicate the presence of non-spherical, depolarizing particles of dust nature.

**References:**

Engelmann, R., Kanitz, T., Baars, H., Heese, B., Althausen, D., Skupin, A., Wandinger, U., Komppula, M., Stachlewska, I. S., Amiridis, V., Marinou, E., Mattis, I., Linné, H., and Ansmann, A.: The automated multiwavelength Raman polarization and water-vapor lidar PollyXT: the neXT generation, Atmos. Meas. Tech., 9, 1767–1784, https://doi.org/10.5194/amt-9-1767-2016, 2016.

Freudenthaler, V., Esselborn, M., Wiegner, M., Heese, B., Tesche, M., Ansmann, A., MüLLER, D., Althausen, D., Wirth, M., Fix, A., Ehret, G., Knippertz, P., Toledano, C., Gasteiger, J., Garhammer, M., and Seefeldner, M.: Depolarization ratio profiling at several wavelengths in pure Saharan dust during SAMUM 2006, Tellus B, 61, 165–179, https://doi.org/10.1111/j.1600-0889.2008.00396.x, 2009.

Freudenthaler, V.: About the effects of polarising optics on lidar signals and the Δ90 calibration, Atmos. Meas. Tech., 9, 4181–4255, https://doi.org/10.5194/amt-9-4181-2016, 2016.

**Line 442: Why would this case be ideal for evaluating Aeolus/ ALADIN as we know Aeolus cannot measure non-spherical dust properly?**

In a complete Cal/Val study, they must be addressed all the factors determining the agreement between spaceborne and ground-based retrievals. It is well-known that the misdetection of the cross-polar return signals by ALADIN results in a degradation of its performance for depolarizing targets (i.e., dust, volcanic ash, cirrus crystals). This is clearly stated between lines 473 and 477 in the submitted text. Therefore, in the test case of 10$^{th}$ July 2019 (Section 6.1.1) we want to demonstrate how much this deficiency affects the comparison between Aeolus and Polly$^{XT}$ retrievals.

**Line 449: What is meant by "statistical uncertainty margin"?**

Below we are providing a short description of the backscatter error in Aeolus and ground-based profiles.

**Aeolus**

We are copying below from the last paragraph in Section 2.3.1 in Flament et al. (2021).

"*Equations have been derived to estimate the impact of the detection noise on measured signals $S_{Ray}$ and $S_{Mie}$ on retrieved $\beta_p$ and $\alpha_p$ values. The derivation of these error estimates is fully explained in Flamant et al. (2021) but is too cumbersome to be reported here. It is based on the assumption that the uncertainty of $S_{Ray}$ and $S_{Mie}$ is purely due to the Poisson counting noise and uses second-order developments. As a consequence, error estimates are valid as long as the level of noise is not too high; otherwise, the approximation introduced by the second-order developments becomes too coarse. The errors estimates do not take into account the impact of atmospheric heterogeneity within the BRC that increases the random noise on the BRC accumulation of observation level $S_{Ray}$ and $S_{Mie}$. It nevertheless remains that they are useful to identify the $\beta_p$ and $\alpha_p$ estimations that are reliable and then give a good idea of their accuracy.*"

**Ground-based lidar**

As explained in Mattis et al. (2016), statistical uncertainty in SCC Level 2 products is calculated either by Monte Carlo method or traditional error propagation. The former was selected in our study for all the considered profiles. In case of elastic lidar signals (as those used herein), the calculation of the uncertainty of particle backscatter coefficient profiles with the Monte Carlo method entails:

  i)      The overall statistical error of pre-processed signals. In case of photon-counting systems this can be evaluated for each photon-counting raw signal range bin as the square root of the corresponding count (D'Amico et al., 2016)
  ii)     the assumed particle lidar ratio value and uncertainty

iii)    the statistical error of the signal within the selected calibration range

*References:*

*D'Amico, G., Amodeo, A., Mattis, I., Freudenthaler, V., and Pappalardo, G.: EARLINET Single Calculus Chain – technical – Part 1: Pre-processing of raw lidar data, Atmos. Meas. Tech., 9, 491–507, https://doi.org/10.5194/amt-9-491-2016, 2016.*

*Mattis, I., D'Amico, G., Baars, H., Amodeo, A., Madonna, F., and Iarlori, M.: EARLINET Single Calculus Chain – technical – Part 2: Calculation of optical products, Atmos. Meas. Tech., 9, 3009–3029, https://doi.org/10.5194/amt-9-3009-2016, 2016.*

We have modified the initial text as follows.

"*The colored dashed lines (Aeolus) and the pink shaded area (Polly$^{XT}$) correspond to the statistical uncertainty margins of the spaceborne (see Section 2.3.1 in Flament et al., (2021)) and the ground-based (D'Amico et al., 2016) retrievals, respectively. Both refer to the photocounting noise following a Poisson distribution.*"

**Line 473: The authors mention "fine particles" but an explanation is missing here; we recommend "until their arrival over...".**

In the previous sentence we are mentioning the "type" (i.e., anthropogenic, biomass) of the fine particles. Both "*until*" and "*till*" have the same meaning here.

**Line 480: We recommend "AOS are mainly attributed ..."; the models seem to be taken, once again at face value here (i.e., they seem to be treated the same as observations, but model species (and their spatial variation) are sometimes not reliable).**

We think that we have already provided sufficient answers of how the models are treated in our analysis. This is clearly stated in our replies as well as in the manuscript.

**Line 484: We recommend "ALADIN reproduces the layer's structure well".**

We would like to thank the reviewer for his/her suggestion but we prefer to keep the sentence as is.

**Line 493-496: This sentence is not clear. The fact that MERRA-2 and CAMS aerosol optical properties and speciation disagree cannot be directly connected to a good or bad performance between models and AERONET. One would need to directly compare aerosol optical properties from MERRA-2 or CAMS to aerosol optical properties from AERONET. Also, the way this is written could make it sound like AERONET provides aerosol species, which it does not. Instead, it measures aerosol optical properties, which can be used to define aerosol types and that can be indirectly translated into aerosol chemical species in certain cases (Kacenelenbogen et al. 2022).**

In our study, we don't give emphasis on the evaluation of the performance of the CAMS and MERRA-2 reanalysis products. We think that this is clear and it is beyond the scope of this paper. We assume that the reviewer is referring to the aerosol speciation and not in aerosol optical depth. For the latter one, the reanalysis and the observed AODs can be compared directly (depending whether AOD observations are assimilated or are independent in the assimilation scheme). The discussion in the manuscript is made in a more generic sense. We agree with the reviewer and we are aware that AERONET observations do not provide information for aerosol species. However, the combination of AERONET intensive and extensive aerosol properties, along with ancillary information (e.g., FLEXPART), can provide useful information (even

though it is not complete). For this case (8$^{th}$ July 2020), MERRA-2 obviously fails to reproduce the AOD levels while the strong contribution from sea-salt particles is not justified by the high AERONET Ångström (1.5-1.6) and FMF (0.75-0.95) values, indicating the predominance of fine particles. Please note that we are referring to a specific case and we are not generalizing our results. A better assessment would require LUT tables of the extinction coefficient (at different wavelengths and RH levels, see Section 3 in Randles et al., (2017)) in order to compare the reanalysis optical properties against those given by AERONET.

In the revised manuscript we have modified this part which reads as follows:

*"On 8$^{th}$ July 2020, the broader area of the Antikythera island was under the impact of moderate-to-high aerosol loads, mainly consisting of organic and sulphate particles, in the western and southern sector of the station, based on CAMS simulated AODs (up to 0.5) (Fig. S12-ii). AERONET measurements, yield UV AODs up to 0.5 and Ångström exponent higher than 1.5 during early afternoon (Fig. S13) whereas the FMF is higher than 0.75 throughout the day (Fig. S14). MERRA-2 AOD patterns (Fig. S12-i) and speciation (strong contribution from marine and sulphate aerosols to the total aerosol load) are different from those of CAMS, without being very consistent with respect to the ground-based sunphotometer observations (Fig. S13, Fig. S14)."*

**Line 502: We recommend "Aeolus performance depends on altitude according to Polly...".**

We believe that our version is better stated.

**Line 554: The reader needs to be reminded which case studies are included here – are those the 46 case studies of line 383?**

Thanks for the comment! In the revised manuscript we are clarifying better this point.

**Line 563: The statement referring to "the contribution of depolarizing particles is quite low based on the ancillary dataset" needs more explanation and needs to be supported by some results.**

Please note that in the previous sentences we are mentioning that it is not possible to apply the conversion from linear to circular optical products for the retrievals acquired at Thessaloniki due to technical issues (related to the polarization purity of the emitted laser beam and the performance of the telescope lenses). Therefore, if we consider this conversion, a significant part of the collocated sample cannot be used in the statistical analysis thus reducing the robustness of our results. In the profiles derived in Athens and Antikythera, after applying the conversion in the ground-based profiles, we see that the differences between total (cross plus co) and Aeolus-like (co) backscatter are negligible due to the presence of spherical particles (the ancillary datasets have been used as indicator for further confirmation).

Here we are presenting the vertically resolved metrics (bias, RMSE) for the unfiltered (as in Fig. 4) and filtered (as in Fig. 5) Aeolus profiles. As reference, we are using the ground-based Aeolus-like backscatter retrievals after applying the conversions (from linear to circular optical products) presented in Paschou et al. (2021). Since depolarization measurements are not available at Thessaloniki, the sample for the statistical analysis contains profiles only from Athens and Antikythera. For the unfiltered Aeolus profiles, there are many similarities (except bins 13-19 for SCA) between the results presented here and those given in the manuscript. On the contrary, there are significant differences for the Aeolus cloud-filtered profiles. However, the number of BRCs considered for the metrics calculation is significantly reduced thus making questionable the robustness of the obtained findings. Drastic reductions appear also for the unfiltered Aeolus profiles. Summarizing, we believe that our decision to present in the manuscript the comparison of Aeolus backscatter retrievals against the total backscatter from ground-based lidars is well supported.

**Unfiltered Aeolus profiles**

[Figure]

**Cloud filtered Aeolus profiles**

[Figure]

[Figure]

(ii-a)        (ii-b)

**Figure 4: The metrics need to be described, like in Abril-Gago et al. (2022). Authors should specify why they show Aeolus/ALADIN profiles that are cloud contaminated (or "unfiltered") in their analysis. Is is not obvious why there would not be more disagreement between cloud-unfiltered space-borne and cloud-filtered ground-based profiles compared to cloud-filtered space and ground profiles. Or is this a way to test their SEVIRI-based cloud filtering method?**

We don't think that we have to provide the formulas for the bias and the root mean square error. These two metrics are two of the most well-known and they have been applied in numerous studies in atmospheric sciences. In the revised manuscript we have added the following reference:

*"Wilks, D.S. Statistical Methods in the Atmospheric Sciences, 4th ed.; Elsevier: Cambridge, MA, USA, 2019."*

Regarding the second part of the reviewer's comment, please see our reply in a previous similar comment raised by the reviewer.

**Line 586: We recommend "Fig. 4" instead of "Fig. 5".**

Done. Thanks for the correction!

**Line 591-592: The authors should mention that "SCA mid-bin" is expected to perform better than SCA.**

It is already mentioned in the text.

**Line 597-599: This is a repeat from line 137.**

In lines 136-138 we are mentioning the work of Abril-Gago et al. (2022) whereas in lines 597-599 we are stating that the findings between these two works are in a very good agreement.

**Line 624: Do the authors mean low SNR instead of high SNR?**

Thanks a lot for noticing our mistake! It has been corrected in the revised manuscript.

**Figure 6-7: Again, why show cloud contaminated Aeolus/ALADIN profiles? Also, why show SCA instead of SCA_bin as the latter is expected to lead to better results (already shown in Fig. 5)?**

Because we want to contrast the results between the two Aeolus vertical scales and between cloud filtered and unfiltered profiles.

**Line 641: Again, can Aeolus/ALADIN profiles still be cloud contaminated after applying the SEVIRI cloud mask?**

This might be possible, but we cannot be sure due to the lack of a classification scheme in Aeolus retrievals.

**Line 641-643: This statement about not using QA flags appears too late in the text. It should be in the method or the Aeolus/ALADIN section.**

We believe that it is already well placed.

**Line 654: "many similarities" needs to be described in more detail.**

We believe that it is pretty clear the meaning of this sentence. The scatterplots in Abril-Gago et al. (2022) and in the current study look very similar.

**Line 682-688: The authors should explain why they expect a difference in performance between the ascending and descending orbital data. Grouping the data per orbit direction seems inconclusive and we question the usefulness of mentioning the results.**

Considering the strong temporal variation of aerosol loads (even at short temporal scales) there could be differences between dawn (early morning) and dusk (early afternoon) orbits. Please note that we are briefly discussing this part and we are mentioning that we need more data (e.g., EARLINET study) in order to derive robust results.

**Line 701-705: This appears too late in the text. It should be part of the comparison method between Aeolus/ALADIN and ground-based lidars.**

We believe that this sentence fits better in Section 7 where we are discussing the Cal/Val aspects and the recommendations.

**Line 702: It is not clear what the authors mean by "the theoretical assumptions".**

The formulas for the conversion from linear to circular optical products are valid under the absence of orientation of the suspended particles and multiple scattering effects.

**Line 719-722: Again, the authors should add some nuance to the discussion on model performance.**

We have rephrased the sentence.

**References:**

**Abril-Gago, Jesús, et al. "Statistical validation of Aeolus L2A particle backscatter coefficient retrievals over ACTRIS/EARLINET stations on the Iberian Peninsula." Atmospheric Chemistry and Physics 22.2 (2022): 1425-1451.**

**Anderson, Theodore L., et al. "Mesoscale variations of tropospheric aerosols." Journal of the Atmospheric Sciences 60.1 (2003): 119-136.**

**Baars, H., Radenz, M., Floutsi, A. A., Engelmann, R., Althausen, D., Heese, B., et al. (2021). Californian wildfire smoke over Europe: A first example of the aerosol observing capabilities of Aeolus compared to ground-based lidar. Geophysical Research Letters, 48, e2020GL092194. https://doi.org/10.1029/2020GL092194**

**Dubovik, Oleg, et al. "Variability of absorption and optical properties of key aerosol types observed in worldwide locations." Journal of the atmospheric sciences 59.3 (2002): 590-608.**

Flament, T., Trapon, D., Lacour, A., Dabas, A., Ehlers, F., and Huber, D.: Aeolus L2A aerosol optical properties product: standard correct algorithm and Mie correct algorithm, Atmos. Meas. Tech., 14, 7851–7871, https://doi.org/10.5194/amt-14-7851-2021, 2021.

Kacenelenbogen, Meloë SF, et al. "Identifying chemical aerosol signatures using optical suborbital observations: how much can optical properties tell us about aerosol composition?." Atmospheric Chemistry and Physics 22.6 (2022): 3713-3742.

Lee, Logan, et al. "Investigation of CATS aerosol products and application toward global diurnal variation of aerosols." Atmospheric Chemistry and Physics 19.19 (2019): 12687-12707.

Russell, Philip B., et al. "A multiparameter aerosol classification method and its application to retrievals from spaceborne polarimetry." Journal of Geophysical Research: Atmospheres 119.16 (2014): 9838-9863.

Sayer, Andrew M., and Kirk D. Knobelspiesse. "How should we aggregate data? Methods accounting for the numerical distributions, with an assessment of aerosol optical depth." Atmospheric Chemistry and Physics 19.23 (2019): 15023-15048.

Shinozuka, Y. and Redemann, J.: Horizontal variability of aerosol optical depth observed during the ARCTAS airborne experiment, Atmos. Chem. Phys., 11, 8489–8495, https://doi.org/10.5194/acp-11-8489-2011, 2011.

We would like to thank the Reviewer for his/her thorough report. Thanks to his/her constructive comments our submitted manuscript has been substantially improved. Below are given point-by-point replies (regular font) to the comments (bold font) raised by the Reviewer.

**The authors perform an assessment of the SCA backscatter coefficient product from the Aeolus satellite by comparison to ground based lidar observations. They have split their work in two parts. First, an analysis of four dedicated, illustrative test cases is presented, including a creditable multitude of ancillary data that provides information on the aerosol origin and type. In a second part, all available collocation cases over the chosen lidar stations contribute to a statistical analysis of bias and RMSE, spanning the current mission lifetime. In lack of a cloud mask within Aeolus' data products, the authors efficiently filter the data themselves and can thereby show moderate to good performance of Aeolus backscatter coefficients. However, the findings suggest that particularly the retrieved backscatter coefficients closest to the ground are not reliable since they suffer from low SNR. The retrieved backscatter coefficient above the ground is biased due to surface reflectance. However, there are some substantial changes and clarifications necessary before publication of the work.**

**General (Major) Comments**

**I agree with referee 1 that the wording and sentence structure throughout the manuscript makes it often more difficult to grasp. That is particularly because of numerous insertions into the sentences, separated by commas or parentheses, and maybe a general trend for nouns over verbs. To provide only one example from L.468 "Under the prevalence of the Etesian winds (Tyrlis and Lelieveld, 2013), a typical pattern dominating over the broader Greek area during summer months, when winds blow mainly from NNE directions, anthropogenic aerosols from megacities (Kanakidou et al., 2011) and particles originating from biomass burning in the eastern Europe and in the surrounding area of the Black Sea (van der Werf et al., 2017) are transported southwards.". The main clause "Under the prevalence of winds [...] aerosols [...] are transported southwards." is stretched out too much.**

We would like to thank the reviewer for his/her comment. We have revised the manuscript trying to "simplify" the text and reduce lengthy (or complicate) sentences.

**Some parts of the manuscript seem not to contribute to or distract from the scope of the paper. Some sections or paragraphs could potentially be shortened or omitted, by asking who the audience of this work is. E.g. the second section with the Aeolus instrument description contains very general information that is mostly not used throughout the rest of the manuscript and can therefore be referenced (see suggestions in specific comments). Also, the conclusion can be made more compact by separating it into a conclusion and an outlook section, or can be condensed in other ways (also see suggestions in specific comments).**

Please see our replies below in the relevant specific comments. As a short note, in most cases we have modified the manuscript as suggested by the reviewer.

**It is good that the authors assess the aerosol climatology via the MODIS-Aqua AODs. However, the performed analysis of concentric circles seems not well suited for the assessment of the horizontal heterogeneity, see specific comments regarding L.351-366.**

We have updated substantially this part of our work. Please see our detailed reply in the relevant specific comment.

**Throughout most of the text, the authors do not differentiate between the performance of the Aeolus satellite itself and the performance of the retrieved SCA co-polar backscatter coefficient within the L2A product. This needs to be clarified, particularly since two significantly improved optical properties products are available as of March this year (see specific comments).**

As it is explained below, we have modified the relevant parts as suggested by the reviewer.

**The currently implemented collocation method appears to me to have an offset of about 45 km in flight direction, since only the start of a BRC but not its center location is used for the distance calculation to the lidar ground stations. If I did not miss something, this will need to be adjusted, making necessary to reanalyse the data and update the corresponding plots.**

We agree with the reviewer that it would be better to use the coordinates of the BRC center instead of the beginning of the satellite scan. Nevertheless, this has an almost negligible impact in Thessaloniki and very small in Antikythera. On the contrary, in Athens, due to the "peculiarity" of the site such decision would exclude most of the matchups between Aeolus and ground-based profiles since ALADIN track resides near the edge of the defined circle. Therefore, we think that it is better to proceed with our initial approach trying not to reduce further the already limited number of cases and BRCs.

In order to illustrate how many BRCs are well spatially collocated with ground-based profiles, we are providing a table of all the considered cases denoting with green boxes the BRCs (either red or blue or magenta; see Fig. 2-iii) where at least half of its length resides within the circle whereas the opposite is displayed with red rectangles. The boxes with X symbol indicate that the corresponding BRCs do not satisfy the spatial collocation criterion. Overall, in 77% of the total number of BRCs (85) there is not any "impact" of which coordinates are used for the spatial collocation.

| Case | Date | Station | Orbit | RED | BLUE | MAGENTA |
|---|---|---|---|---|---|---|
| 1 | 06/11/2019 | ATHENS | Dawn | [RED] | X | X |
| 2 | 18/12/2019 | ATHENS | Dawn | [GREEN] | [RED] | X |
| 3 | 15/01/2020 | ATHENS | Dawn | [RED] | X | X |
| 4 | 22/01/2020 | ATHENS | Dawn | [GREEN] | X | X |
| 5 | 13/05/2020 | ATHENS | Dawn | [RED] | X | X |
| 6 | 20/05/2020 | ATHENS | Dawn | [GREEN] | X | X |
| 7 | 01/07/2020 | ATHENS | Dawn | [GREEN] | [RED] | X |
| 8 | 15/07/2020 | ATHENS | Dawn | [GREEN] | [RED] | X |
| 9 | 22/07/2020 | ATHENS | Dawn | [RED] | X | X |
| 10 | 29/07/2020 | ATHENS | Dawn | [GREEN] | X | X |
| 11 | 09/09/2020 | ATHENS | Dawn | [RED] | X | X |
| 12 | 30/09/2020 | ATHENS | Dawn | [GREEN] | [RED] | X |
| 13 | 03/07/2019 | ANTIKYTHERA | Dawn | [GREEN] | [GREEN] | X |
| 14 | 03/07/2019 | ANTIKYTHERA | Dusk | [GREEN] | [RED] | X |
| 15 | 10/07/2019 | ANTIKYTHERA | Dawn | [GREEN] | [RED] | X |
| 16 | 10/07/2019 | ANTIKYTHERA | Dusk | [GREEN] | [GREEN] | [RED] |
| 17 | 17/07/2019 | ANTIKYTHERA | Dusk | [GREEN] | [GREEN] | X |
| 18 | 24/07/2019 | ANTIKYTHERA | Dusk | [GREEN] | [GREEN] | X |
| 19 | 08/07/2020 | ANTIKYTHERA | Dusk | [GREEN] | [GREEN] | X |
| 20 | 29/07/2020 | ANTIKYTHERA | Dawn | [GREEN] | X | X |
| 21 | 05/08/2020 | ANTIKYTHERA | Dawn | [GREEN] | [GREEN] | [RED] |
| 22 | 05/08/2020 | ANTIKYTHERA | Dusk | [GREEN] | [RED] | X |
| 23 | 12/08/2020 | ANTIKYTHERA | Dawn | [GREEN] | [RED] | X |

| # | Date | Location | Time | Col 1 | Col 2 | Col 3 |
|---|---|---|---|---|---|---|
| 24 | 02/09/2020 | ANTIKYTHERA | Dawn | green | X | X |
| 25 | 16/09/2020 | ANTIKYTHERA | Dusk | green | green | X |
| 26 | 23/09/2020 | ANTIKYTHERA | Dusk | green | X | X |
| 27 | 24/02/2021 | ANTIKYTHERA | Dusk | green | green | X |
| 28 | 03/07/2019 | THESSALONIKI | Dawn | green | green | green |
| 29 | 10/07/2019 | THESSALONIKI | Dusk | green | X | X |
| 30 | 24/07/2019 | THESSALONIKI | Dawn | green | green | green |
| 31 | 07/08/2019 | THESSALONIKI | Dawn | green | green | green |
| 32 | 04/09/2019 | THESSALONIKI | Dawn | green | green | red |
| 33 | 18/09/2019 | THESSALONIKI | Dusk | green | X | X |
| 34 | 16/10/2019 | THESSALONIKI | Dusk | red | X | X |
| 35 | 23/10/2019 | THESSALONIKI | Dawn | green | green | green |
| 36 | 08/01/2020 | THESSALONIKI | Dawn | green | green | green |
| 37 | 15/01/2020 | THESSALONIKI | Dawn | green | green | X |
| 38 | 08/04/2020 | THESSALONIKI | Dawn | green | green | X |
| 39 | 06/05/2020 | THESSALONIKI | Dawn | green | green | red |
| 40 | 13/05/2020 | THESSALONIKI | Dawn | green | green | red |
| 41 | 10/06/2020 | THESSALONIKI | Dawn | green | green | green |
| 42 | 01/07/2020 | THESSALONIKI | Dawn | green | green | green |
| 43 | 22/07/2020 | THESSALONIKI | Dawn | green | green | red |

**Subsections 6.1.3 and 6.1.4: In my opinion, the descriptions and conclusions of the individual Aeolus lidar profiles in Fig. 3 may be much to detailed and flawed. I explain in my specific comment on L.502-503, that there is reason to believe that the discussed discrepancies are just noise induced and therefore the reached conclusions are not valuable or generalizable. I recommend the following procedure: As a first validation step, I encourage the authors to provide Figure 3 with all negative SCA backscatter values shown. This will provide an impression of the actual noise level encountered in the SCA backscatter in the different test cases. I expect to see values up to minus 0.5-1 Mm-1sr-1 in some cases in accordance with e.g. Fig. 8 in Ehlers et al. (2022, doi.org/10.5194/amt-15-185-2022). If that is indeed the case, then the discrepancies along the profiles may be mostly noise induced and the current, detailed conclusions must be reconsidered, i.e. the authors should test for the hypothesis and make accordingly changes to the text. In this case, especially the statement in the abstract L.41-43 "The level of agreement between spaceborne and ground-based retrievals varies with altitude when aerosol layers, composed of particles of different origin, are stratified (8th July 2020, 5th August 2020)." is contestable.**

We have reproduced the plots of Figure 3 by decreasing the lower limit of x axis down to -1 Mm$^{-1}$sr$^{-1}$ thus visualizing negative backscatter coefficients. Following the reviewer's suggestion, we have modified accordingly the discussion in Sections 6.1.3 and 6.1.4 in the revised text.

[Figure]

**Specific comments**

**L.38-43 This could be more compact, considering it is in the abstract. Particularly the discussion of the 4 test cases seems very specific and could be condensed into a shorter sentence.**

We believe that this part is already short and compact. In the revised manuscript, we have slightly modified the text explaining that our results refer to specific cases. This is done in order to avoid any possible confusion that these findings can be "generalized" for the entire Aeolus L2A dataset.

**L.41-43 The "level of agreement" is not strictly defined and, hence, seems subjective. In my opinion, this statement is presented too confident. In fact, this is concluded from two single BRC of Aeolus. The authors themselves stress the issues with collocation, so I am not convinced at this point that the remaining variations in the profiles are caused only by the stratification, but can originate from horizontal inhomogeneity of the atmosphere's aerosol load. The supplementary material helps only little, since the models provide AOD only.**

We have changed this sentence in the revised text.

"*For the rest two cases (8th July 2020, 5th August 2020), due to noise issues, Aeolus performance downgrades in terms of depicting the stratification of aerosol layers composed of particles of different origin.*"

**L.74-80 This sentence is too long and the last part seems not to fit in gramatically.**

We have rephrased the sentence as follows:

"*Therefore, this deficiency hampers a reliable quantification of the suspended particles' load within the planetary boundary layer (PBL), related to health impacts. Moreover, it is not feasible to depict the three-dimensional structure of transported loads in the free troposphere, linked to aerosol-cloud-radiation interactions and associated impacts on atmospheric dynamics (Perez et al., 2006; Gkikas et al., 2018; Haywood et al., 2021). Likewise, passive aerosol observations are not suitable for monitoring stratospheric long-lived plumes that affect aerosol-chemistry interactions and perturb the radiation fields (Solomon et al., 2022).*"

**L.82 "as well as the geometric features of the particle's layers" What are the "geometric features", if not the already-mentioned vertical structure? Please omit or specify.**

The geometric features of the particle layers and the vertical (or three dimensional) structure of the aerosol layers have the same meaning. We think that we can use them in the text without confusing the reader.

**L.94-115 This paragraph provides an overview of the L2B wind product development and application. Considering the paper's scope of aerosol backscatter assessment, I recommend to omit/condense it.**

We have reduced the length of this paragraph.

"*On 22nd August 2018, the European Space Agency (ESA) launched its Earth Explorer wind mission, Aeolus. It is the first space-based Doppler Wind lidar worldwide, and was a major step forward for Earth Observations (EO) and atmospheric sciences. The key scientific objective of Aeolus is to improve numerical weather forecasts and to improve our understanding of atmospheric dynamics and their associated impacts on climate (Stoffelen et al., 2005; Isaksen and Rennie, 2019; Rennie and Isaksen, 2019). After about 1.5 years of instrument and algorithm improvements, the Aeolus L2B wind product was of such good quality that the European Centre for Medium Range Forecasts (ECMWF) could start operational assimilation (January 2020). In May 2020, three further European weather forecast institutes (DWD, Météo-France and the UK MetOffice) started the operational*"

*assimilation of Aeolus winds. All meteorological institutes reported that Aeolus winds had significant positive impact on the short and medium term forecasts, with the largest impact in remote areas less covered by other direct wind observations including the tropics, southern hemisphere and polar areas (e.g. ECMWF 2020; Rennie et al., 2021).*"

**L.116-128 This paragraph might be better placed in / merged with the second section about the ALADIN instrument.**

We agree with the reviewer and we have modified the text as suggested.

**L.169, Section 2; While reading this section I was reminded of other Aeolus related works. In fact, I was wondering whether the degree of detail is relevant for the audience of your manuscript, or if you could get away with a more high-level description of the Aeolus typical vocabulary only (as in the L2a user guide). Essentially, the information here can be looked up in the Aeolus Science Report or many other papers. But since your work focuses on validating the data rather than e.g. modifying the L2A processing chain or including so-far unknown instrumental effects, it may be a consideration to omit most parts for brevity.**

We have reduced Section 2 in the revised manuscript.

**L.192-194 It is unclear with which property the angle increases. Please make the formulation unambiguous by changing the statement to something like e.g. "The 35 degree off-nadir pointing corresponds to an angle of about 37.6 degree with the Earth surface, due to its curvature".**

We have adjusted the manuscript according to the reviewer's suggestion.

**L.205, Section 3; This description of the L2A data product is outdated at least with the start of the new baseline 2A14 from 29th March 2022. It has been decided to remove the ICA product completely and two new optical property products have been added, namely the SCA-MLE Optical Properties (Ehlers et al., 2021) and the AEL-PRO Optical Properties (from adjusted EARTHCARE algorithms). Both products are expected to bring considerable improvement over the SCA, since the inverse retrieval problem is solved not algebraically but via state-of-the-art methods (Maximum Likelihood Estimation, Optimal Estimation, respectively), see Ehlers et al. (2021) for the SCA-MLE product. These changes are tracked e.g. in the Aeolus Level 2a Processor Input/Output Data Definition available here: https://earth.esa.int/eogateway/documents/20142/37627/Aeolus-L2A-Input-Output-Data-Definitions-Interface-Control-Document Please give an adequate description of the data product, in order to put your analysis in the correct context. To my knowledge the Aeolus mission data has not yet been reprocessed with the new processors, which then offers potential for future studies.**

We have updated Section 3 in the revised manuscript as suggested by the reviewer.

**L.238 NITWT is not the name of the method, but the name of the variable that allows for simpler notation.**

Thanks a lot for the correction.

**L.243-250 This paragraph is a perfect introduction to then mention the SCA-MLE and AEL-PRO optical properties data products, which aim to mitigate such problems to a big part. A brief description could be added hereafter to update the section. It must be stressed that also the backscatter profits from the processing update!**

We have followed the reviewer's recommendation.

**L.250-252 This is not a primary reference for the zero-flooring. The primary reference is Flament et al. (2021) or the L2A Algorithm Theoretical Baseline Document, section 6.2.2.1, see here: https://earth.esa.int/eogateway/documents/20142/37627/Aeolus-L2A-Algorithm-Theoretical-Baseline-Document.**

Thanks a lot for the correction

**L. 351-366, Criterion for spatial homogeneity; The authors want a measure for spatial homogeneity of the atmosphere's aerosol load on an instantaneous base. However, the presented, concentric, climatological analysis is not suited for these needs for at least two reasons. Also, the description lacks some detail. The two main points below:**

**i) Figure 1 provides one AOD value per concentric circle and location. So the reader has to assume that in addition to the spatial average a temporal average over the 10 year period has been performed. This average is not mentioned in the text and the word "climatological" appears only in the figure caption. Therefore, the word "Annual" in Fig. 1i and 1ii is misleading.**

In the revised manuscript we are clarifying better the averaging procedure both in spatial and temporal terms.

**Now, averaging the AOD pattern over time will potentially smoothen out most of the horizontal heterogeneity of AOD that is present on a daily basis. However, the latter is the desired property in order to assess the quality of collocation. An (oversimplified) counter example goes as follows: Assume AOD pixels follow a chess board pattern (with changing locations over time due to wind). This would show a lot of heterogeneity, hampering collocation. But due to the two averages, one over the rings and one over time, the developed criterion would indicate perfect homogeneity.**

We can understand the point raised by the reviewer. Nevertheless, the AOD pattern around a station cannot have a chess board structure. Depending on the station location and the prevailing meteorological/aerosol conditions the AOD in the vicinity of the station has "specific" spatial patterns, which can vary in temporal terms. Among the selected stations in the current study there is a clear contrast between Antikythera (background aerosol conditions) and Athens/Thessaloniki (urban aerosol conditions). This is quite evident in the urban sites where the AOD decreases rapidly for increasing radii. A critical point mentioned by the reviewer is the AOD variability in time and we admit that this aspect has not been appropriately treated in the submitted manuscript. In the revised text, we are presenting the coefficient of variation (CV) defined as the ratio of the standard deviation and the arithmetic mean (Anderson et al., 2003). CV expresses how much variable is the AOD, with respect to its mean value, in temporal terms. For completeness, we have also calculated the spatial autocorrelation (the correlation matrices are given in the revised supplement) among all the possible combinations of the defined circles. Since we are processing the MODIS swath data (they are not provided on a gridded structure) and we are selecting only AOD retrievals of best quality (QA=3) (many AODs have been discarded) we believe that it is better to work with the daily spatial AOD averages of each circle.

**ii) Another shortcoming is the possibly very location specific outcome of this analysis: In my opinion, there is no reason to either favour Aeolus' frequent observation location or the ground-based lidar location as a center for the concentric circles. However, if the center was chosen e.g. 80 km away from pollution sources such as Athens or Thessaloniki, then their increased AOD pixels would be averaged with all pixels from a whole ring of unrelated locations, including ocean, meadows and villages 160 km away. This way, pollution sources will be hidden by averaging, if the circles' center is not coincident with them, making the presented analysis little robust.**

**Standard tools such as 2D autocorrelation functions of the AOD "images" would not suffer from such shortcomings.**

We believe that the updated analysis addresses all the necessary aspects regarding the horizontal AOD variability in the vicinity of the PANACEA sites. Our spatial collocation criterion is the common procedure applied in numerous studies related to the evaluation of satellite retrievals. The station must be the center of the circle (or square) area. For the 2D autocorrelation, we think that the reviewer assumes that the MODIS

AOD data are provided in a gridded structure. However, this is not the case for the MODIS L2 AOD covering the swath sampled area by the satellite (5-min segment).

**L.374 You take the beginning of the scan as the location of an Aeolus BRC, however, its middle is more representative as centerpoint of the measurement but lies about 45 km further away. When considering Figure 2(i), imagine now that by random chance, Aeolus had started scanning each BRC 5 km earlier. Then the BRC that is now red would not be considered in the analysis at all, though still closest. This means, your collocation criterion is currently offset by about 45 km in flight direction, which is quite a lot! This must be fixed and can be done, e.g. approximately, by applying the running average filter ([0.5 0.5]) over the current latitude and longitude arrays, or by assuming the satellites' speed and direction. Otherwise, the location of the center measurement within the BRC can be extracted from elsewhere in the L2A product, to the best of my knowledge.**

Please see our reply above in the general comments.

**Fig.2 The tips of the orange arrows are barely visible, please enhance.**

We have changed the color of the arrows.

**L.380-383 The collocation criteria should be objective, so please quantify by up to how much time they have been relaxed, another hour? 2 hours?**

We have revised this part of the manuscript providing more details. Below is given the modified text.

*"For the ground-based observations, the aerosol backscatter profiles are derived considering a time window of ± 1 hour around the satellite overpass. Nevertheless, this temporal collocation criterion has been relaxed or shifted in few cases to improve the quality of the ground-based retrievals (i.e., by increasing the signal-to-noise ratio) as well as to increase the matched pairs with Aeolus L2A profiles. Both compromises are applied since the weather conditions favoring the development of persistent clouds may eliminate the number of simultaneous cases. It is noted, however, when the temporal window is shifted or relaxed we are taking into account the homogeneity of the atmospheric scene (probed by the ground lidar). For the Antikythera station we did not deviate from the pre-defined temporal criterion apart from one case study. In Thessaloniki and Athens, the time departure between Aeolus and ground-based profiles can vary from 1.5 to 2.5 hours. Overall, 43 cases are analyzed out of which 15 have been identified over Antikythera, 12 in Athens and the remaining 16 in Thessaloniki."*

**L.395-397 When reading the manuscript, my burning question was, how many of the above mentioned cases/BRCs remained after filtering. I only found this information much later in the text. Could the authors please consider moving this information up here?**

Please note that we are discarding cloud contaminated BRCs and not cases (i.e., days). We think that it is better to discuss the reduction of the BRCs in Section 6.2 trying to avoid any confusion to the reader.

**L.408-410 Can the authors motivate here why explicitly these cases where chosen? Also, I wonder which criterion was applied to choose a single BRC out of each case, presented in Fig. 3. The spatially closest? The visually most representative? I presume that at least for one of the cases there was more than one BRC to consider.**

There is not any specific criterion. These were the most interesting cases, typical in the eastern Mediterranean, from our collocated sample. Regarding the BRC, we are selecting the nearest one to the station coordinates that falls entire within the circle area.

**L.442-444 The word "ideal" is exaggerated.**

We have replaced "ideal" with "appropriate".

**L. 484-486 As the authors report themselves earlier, the backscatter coefficient in SCA and SCA midbin is essentially identical, just averaged onto two different scales. Hence, I do not support this argument of overestimation/underestimation and find it misleading. Also, with a quick look it seems that both are overestimating. Do the authors mean that the layer reaches too far up in SCA midbin? Please specify.**

We have kept the word "overestimates" and we have removed the second parenthesis which might cause a confusion.

"*Under these conditions, ALADIN is capable of reproducing satisfactorily the layer's structure (SCA retrievals - brown curve) whereas slightly overestimates its intensity with respect to the ground-truth retrievals.*"

**L.502-503 This sentence is stated with a suggestion, while in fact the information should just be that Aeolus and PollyXT do not agree over the entire profile, and where. As you are well aware this mismatch can have various reasons but a lack of performance of Aeolus.**

Below we are providing the rephrased sentence.

"*For this specific case, Aeolus' performance reveals an altitude dependency according to the comparison versus Polly$^{XT}$ vertically resolved retrievals.*"

In the following sentences, we are describing in detail the Aeolus-Polly$^{XT}$ comparison results throughout the profile.

**At this point I want to also mention, that the error bars on the L2A products are not found to be accurate, and hence suggest a wrong sense of precision, see the recent work of Adrien Lacour from Meteo France and e.g. Fig. 8 in Ehlers et al. (2022): In this test case, the MLE retrieval brings the optical properties much closer to the ground truth than SCA and SCA midbin. So the gaping disagreement between the SCA or SCA midbin and the ground truth is apparently due to noise in the cross-talk corrected particulate (Y) and molecular (X) signals (beta_p=Y/X*beta_m). The true magnitude of this noise can also be illustrated with the magnitude of the negative backscatter values in almost clear atmosphere, which unfortunately are not shown in Figure 3. However, Figure 6 gives a good idea of the spread of negative values in SCA, indicating a ballpark value of 0.5 up to 1 Mm-1sr-1 around the GROUND beta = 0.**

We have reproduced the plots in Figure 3 showing the negative values as suggested by the reviewer and we have updated the relevant discussion in the manuscript.

**Now, the discussion following in L.502 to 507 focuses entirely on the value of two Aeolus bins between 2 and 4 km altitude, the errors of which are most likely underestimated. However, the discrepancy with the ground truth is not much bigger than the approximate noise amplitude estimated from Figure 6 above. Therefore, it is very much possible that these discrepancies in just these 2 bins are indeed caused by noise! Therefore, it is not reasonable to generalize from these results an altitude dependent performance and to conclude a contradiction between the observations on such a weak fundament. This can only be done statistically.**

The discussion between lines 502 and 507 focuses on the aerosol layers found between 6-8km and 2-4km. We have rephrased this part of the text. We think that in Section 6.1 it is clear that we are discussing each case individually and we are not "generalizing" our results. This is done in the statistical analysis presented in Section 6.2.

**L.530-535 I see no to little reason to underline that SCA midbin is "better" in this particualar case. As you point out, SCA midbin has just worse resolution, which helps to reduce the mismatch here, because averaging consecutive bins also reduces the noise. This is no characteristic of this particular lidar profile**

**but follows from the math: In general, SCA midbin is worse at high SNR because of the resolution loss, but appears better in low SNR due to the additional smoothing that the average implies.**

We have modified the text accordingly.

**L.576-577 Can you specify how the ground profiles have been rescaled to match vertically the Aeolus bins? Were the ground-based observations averaged onto both different scales or simply sub-sampled? The latter is not preferred. Please provide an explanation or formula.**

Thanks a lot for noticing our shortcoming. In the revised text we are clarifying how the rescaling is done.

**L.579-585 The range bin index is a tricky reference to perform the analysis on, but I see the need for this implementation. However, it should be stressed in a separate sentence that, this way, one may mix up bins of e.g. 250 m size with bins of 1 km size, which have different noise properties. This is important for interpreting the reliability of RMSE and bias.**

We agree with the reviewer but this is already mentioned in the submitted text (lines 579 – 581) as well as in Section 2.

**L.586-596 Reading the text while looking at the figure, I cannot follow the choice of the authors to discuss the groups of bins 1-3, 4-12 and 12-23 separately. Can the motivation be explained? The bias and RMSE within the group 4-12 is anything but homogeneous.**

The reason is that we are "defining" these groups based on the altitude (given into the parentheses) within the atmosphere. The first three bins reside within the PBL, from bin 4 to 12 we are in the free troposphere and the highest bins the upper/lower troposphere/stratosphere.

**L.593 "the most important finding is that Aeolus is not capable to reproduce satisfactorily the backscatter profiles" I find this a bolt statement to make here. It is not Aeolus but specifically the current Aeulos SCA product in absence of cloud flagging. The cloud-flagged observations, presented some lines thereafter, let you draw a very different conclusion!**

This is exactly what we want to show here and we believe that it is clearly stated in Section 6.2.1. In Figure 4, we are presenting the evaluation metrics for the Aeolus SCA raw (aerosols plus clouds) products and in Figure 5 the corresponding results for the SCA Aeolus cloud-filtered retrievals. Through this comparison it is highlighted the necessity of removing cloud-contaminated Aeolus profiles when compared with ground-based cloud-free retrievals. To summarize, we do not see which is the confusion here.

**L.600-603, point ii); It is not motivated how increased noise causes bias in backscatter coefficients (I assume that "overestimation" is used synonymously to bias, if so, please use "bias" throughout the manuscript whenever appropriate). This is explained in Sec. 4.1 & 4.2 in Ehlers et al. (2022), so maybe reference here as well?**

Done. Thanks a lot! We think that there is not any confusion between overestimation and bias. Both have the same meaning.

**L.615 It should be already explained here, that this low positive bias is due to omitted negative backscatter values (this can be seen in the scatterplots), as you do later in L.662-665. To my knowledge, the corresponding L2A processor parameter has been adjusted so that negative backscatter values in SCA midbin are not just omitted in the newer baselines!**

Done.

**L.622-625 It should be made clear that this statement regards only the bin closest to the surface. Also, "level" should be replaced by "bias".**

In the revised text we are specifying that these bins are close to the surface. We think that it is not correct to use the word bias here since we are discussing the RMSE levels.

**L.644-646 I would not use the word contradiction. I can simply not be said based on metric 1,2 and 3 which product is "better". However, it should also be mentioned that the SCA midbin scatterplot contains less data due to the inherent flagging of negative values, see scatterplots, and hence the analyses are not strictly comparable. Also, the discussion whether SCA backscatter or SCA mb backscatter is "better" depends simply on SNR, as has been addressed in my comment on L.530.**

We have modified the text accordingly.

**L.672 please use "bias" as in Table 1, instead of "overestimation", in order not to confuse the reader whether or not these are two different statistical properties.**

We have replaced the word as suggested by the reviewer.

**Table 1&2; provide units!**

Done.

**L.668-678 This paragraph describes the statistics of the unfiltered data in detail. At this point, it has already been made clear to the reader that the unfiltered data is not suited for statistical analysis. Hence, Table 1 may as well be moved into the Appendix and may be kept for the sake of completeness, including a hint in the text. Instead, a bit more detail about the metrics for the filtered backscatter profiles would be appropriate in L.678-681.**

We prefer to keep the initial version of the text and the Table 1 as is. We believe that our description is well stated.

**L.722 What does "they" refer to?**

We have rephrased the sentence as follows:

"*Over areas with a complex terrain, vertical inconsistencies between ground-based and satellite profiles (reported above ground where its height is defined with respect to the WGS 84 ellipsoid), not physically explained, can be recorded.*"

**L.748 It's rather "SCA backscatter coefficients" to be specific.**

Done.

**L.763 Please specify and write "Aeolus' SCA backscatter product" instead of "Aeolus".**

Done.

**L.781-790 Partly repeats the cross-polar misdetection mentioned above in L.757. Also, this paragraph does not contain a conclusion from your analysis seems detached. It resembles more of an Aeolus-2 future mission outlook? Maybe move into a separate section "outlook", if the information is crucial in your opinion?**

We have removed the whole paragraph in the revised manuscript.

**L. 793-797 The content of this text should be moved into Section 3, since these products have been released already at the end of March 2022. Specifically, it needs to be clarified that not only the extinction but also the quality of the backscatter coefficients (especially precision) is significantly increased with the Maximum-Likelihood Estimation (MLE), making new Cal-Val studies worthwhile once there processed data is available.**

We agree with the reviewer and we have modified the manuscript according to his/her comments.

**L.804-810 This also reads as a mission outlook rather than as a part of your conclusion and may be dropped or moved into a separate section "outlook".**

We prefer to keep it as is since in the last paragraph we are discussing the ongoing and future Aeolus related activities.

**Technical corrections**

**L.230 Refer to the "C coefficients" as cross-talk coefficients as above. In general, using words as "so-called" and setting words in quotation marks should be avoided. It suggests little reliability.**

Done.

**L.241 "downwards", same comment as in L.230.**

Done.

**L.343 The formulation seems odd. Just write "in Section 5" and omit the part in parentheses.**

Done.

**L.369-370 This sentence is wordy/bulky. Better: "The Aeolus L2A backscatter profiles are compared to the measurements of three PANACEA lidar stations."**

We think that our version is better than those suggested by the reviewer.

**L.384 replace "rest" by "remaining".**

Thanks!

**L.481-483 The information in the parentheses is different from the information in the text (SCA vs. Ground and Ground vs. Aeolus-like Ground observations).**

We don't see any mistake in this sentence. Probably there is a misunderstanding with the order of the colors in the parenthesis. In the revised manuscript we are mentioning first the pink (total backscatter) and then the blue (Aeolus-like).

**L.586 This should be Fig. 4 instead of 5.**

Thanks for the correction!

**L.653 replace "not any" with "no".**

Done.

**Ref list: Ehlers et al. (2022) is not included though cited in the text?**

Thanks a lot for noticing our shortcoming.

We would like to thank the Reviewer for his/her comments and suggestions. Our replies (regular font) for each comment (bold font) are provided below.

**Reviewer #3**

**Gkikas et al. compared the Aeolus L2A particle backscatter coefficient retrievals with ground-based lidar measurement in Greece. The authors showed Aeolus SCA and GRD backscatter profiles for 4 cases and statistic assessments for 46 collocated cases. It is not clear if the 4 cases are representative for the SCA backscatter coefficient product. For the statistic assessment, the authors showed that the SCA (SCA mid bin) cloud filtered backscatter profiles have better agreement with the GRD backscatter profile than the unfiltered profiles. The authors used AERONET, CAMS, MERRA-2 aerosol data to describe the aerosol situations for the 4 cases but not compared AOT from the auxiliary data with L2A. It may give readers a feeling that more auxiliary information than the Aeolus L2A data is used in the paper. The paper is well-written, good structure and lots of references. Some long sentences can be rewritten to make the paper easy to read.**

The four cases presented in Section 6.1 correspond to some typical aerosol conditions in the E. Mediterranean under the prevalence of different aerosol species in the broader area of the Antikythera island. Unfortunately, due to our relatively small sample it is not feasible to include more cases. Nevertheless, we are collecting ground-based measurements during Aeolus overpasses and we hope that we will identify new interesting cases. We agree with the reviewer that there is a confusion to the reader as it is written in the submitted text. We are clarifying better this point in the revised manuscript. The utilization of several ancillary datasets is necessary in order to characterize the probed atmospheric scene since there is not this capability on Aeolus retrievals. The comparison between Aeolus AOD against those provided by the ancillary observations/outputs cannot be made at this phase due to the very noisy extinction profiles. Finally, in the revised text we have reduced the long sentences thus simplifying the readability of the manuscript.

**Specific comments**

**Abstract**

**Line 27 Change 'hydrometeors' to clouds. I think hydrometeor is too broad here.**

We changed "hydrometeors" with "clouds" as suggested by the reviewer.

**Please provide the L2A data version (Baseline) in the abstract, because there are different L2A versions available.**

We have added the baseline in the abstract.

**It would be nice to provide some numbers in the abstract.**

We have added few evaluation metrics in the abstract.

**Introduction**

**It is impressive that the authors have cited so many papers throughout the manuscript.**

Thanks!

**Line 285: 'lat = 35.86 N, lon-23.31 E'. The degree symbol is missing. Please check the texts with 'lat=, lon = ' throughout the manuscript.**

Done.

**Line 307: '...at 354 and 532 nm…' Is it 354 or 355 nm?**

It is 355nm. We have corrected it in the revised manuscript.

**Sect. 5 collocation between Aeolus and ground-based lidars.**

**It is not clear how the Aeolus and ground-based lidar are matched in altitude bins. Could you explain it in the texts?**

We are calculating the average value of the ground-based retrievals residing within the Aeolus bin ranges. We are clarifying this point in the revised manuscript.

**Sect. 6.1 results, Please explain why these 4 cases are selected. Are they the best cases?**

In the revised text we are explaining the reason for presenting these four cases.

**Sect. 6.2, Lines 554 – 555: Please move this sentence to the earlier section. It is important to know the L2A data version.**

Done.

**Lines 576-577: '… the GRD profiles have been rescaled to match Aeolus vertical product resolution'. How is the rescaling performed? How many Aeolus profiles are used in the statistic assessment? Later I saw it is in the figures.**

In the revised text we are explaining the rescaling method as well as the number of Aeolus profiles used in the statistical assessment.

**Lines 672 -673: Units are missing after the values.**

Done. Thanks!

**Line 796: '… and the EarthCARE derived AEOL-FF and …' Change AEOL-FF to AEL-FM.**

Thanks a lot for the correction!

---

## Referee Report (RR1)

**General reply**

I would like to cordially thank the authors for their efforts and their detailed response. They made a great effort to restructure parts of the manuscript and to clarify the open questions. The earlier concerns on the cilmatological AOD analysis can be dropped based on the provided explanations and the additional data in the manuscript and supplement (coefficient of variation, spatial correlation and textual changes). Also, the textual changes regarding the 4 specific cases are appropriate, now being less conclusive and leaving open the possibility to noise-induced mismatches between SCA retrievals. I was particularly glad to see the negative values included in the Figure 3, which complement this discussion.
However, two of the raised points have not been addressed to full satisfaction, but both of which will require mainly changes within the text. Hence, I recommend a minor review.

**Point 1: Regarding the colocation criterion**

Thanks for agreeing that it would be better to use the BRC center instead of the BRC start for colocation. I want to thank the authors for the effort of giving a detailed overview of the number of BRCs that are not affected: It is good to know that 77% of the BRCs (observations) are correctly colocated, which makes me confident that the reached conclusions are not too sensitive to any changes. However, below a discussion of open points:

"*On the contrary, in Athens, due to the "peculiarity" of the site such decision would exclude most of the matchups between Aeolus and ground-based profiles since ALADIN track resides near the edge of the defined circle.*"
I disagree with the statement that a change from the starting coordinate to the center coordinate will effectively reduce the number of BRCs.
To illustrate this point, a small graph below with some randomly simulated overpasses (Fig 1): Panel A shows the colocation method as is, illustrating the skewness around the circle in flight direction because of using the starting coordinate. Panel B resembles the procedure that the authors have shown in their rebuttal: Using the starting coordinate for colocation in the first place, but then checking subsequently whether also the center falls inside the circle. This method of course reduces the number of BRCs, because both the center and the starting coordinate are required to fall within the circle. In line with what the authors state, about 41/53 = 77% of the observations were correctly colocated in this example and remain. Nevertheless, panel C shows what happens using the center coordinate right away, which results in a non-skewed choice of BRCs around the circles center.
On average, the version A and C will result in the same number of BRCs, because the density of observations is not reduced, just different BRCs are considered. However, in this random realisation, there are of course small fluctuations allowing for small differences (53 vs 52).

"*Therefore, we think that it is better to proceed with our initial approach trying not to reduce further the already limited number of cases and BRCs.*"

Following my point above, this cannot be the motivation for keeping the flawed colocation method, as the number of BRCs is not reduced by changing to the center coordinates (since other BRCs that were previously disregarded are "entering the game" from below). **Nevertheless, since the 77 % of the BRCs are correctly colocated, I do not expect a huge impact on the statistical analysis and the reached conclusions. So, although I**

**am not agreeing with the authors, redoing the analysis with the correct colocation is probably a disproportional effort compared to the potential gain. I strongly recommend however, that since the authors conclude only satisfactory performance of Aeolus, they should stress clearly the weakness of their applied colocation criterion in the methods section. A reader might argue that any poor performance might be caused by the ~45 km offset, which allows in fact observations at 165 km distance instead of 120 km.**

[Figure]

Fig 1: Illustration of the differences between colocation methods on a random realisation of overpasses (the same realisation is used for panels A, B, C). The panels show the results of different colocation strategies. The satellite is passing from south to north in this example. The color-code follows the Figure 2 of the manuscript.

**Point 2: Differentiating between "Aeolus", the "L2a product" and the "SCA product (or retrieval)"**

I earlier made the remarks:
"Throughout most of the text, the authors do not differentiate between the performance of the Aeolus satellite itself and the performance of the retrieved SCA co-polar backscatter coefficient within the L2A product. This needs to be clarified, particularly since two significantly improved optical properties products are available as of March this year (see specific comments)."
and
">*the most important finding is that Aeolus is not capable to reproduce satisfactorily the backscatter profiles*< I find this a bolt statement to make here. It is not Aeolus but specifically the current Aeulos SCA product in absence of cloud flagging. "

To which the authors answered:

*"As it is explained below, we have modified the relevant parts as suggested by the reviewer."*

Overall, this has not been adjusted throughout the manuscript so I explain below in a more detailed manner: What I meant to stress is that sentences like
*"The most important finding is that **Aeolus** is not capable to reproduce satisfactorily the backscatter profiles."* (L.945 , manuscript with tracked changes)
are not correct. Such statement suggests and concludes incapability of the Aeolus mission itself, while it is particularly the shortcomings of the SCA product that the authors want to underline. Hence, the above statement should be changed to either

*"The most important finding is that **Aeolus' SCA product** is not capable to reproduce satisfactorily the backscatter profiles."* or
*"The most important finding is that **the SCA retrieval** is not capable to reproduce satisfactorily the backscatter profiles."*
I stress this point particularly, since the science teams are and have been working on more refined L2a backscatter coefficient profiles that are already released and included into the L2a product. Hence, there is a need for specification. This is where my impression originated from, that the authors generalize a lot in sections 6.1.x, though I am now sure that this is certainly not intended and hence a misunderstanding due to choice of words. I strongly encourage the authors to differentiate, **so all statements that state "L2a product" and "Aeolus" in place of "SCA product" need to be changed accordingly.**
To provide an extensive list:
L. 39 "L2A backscatter coefficient" → change to "SCA backscatter coefficient"
L. 165, 950 "L2A optical properties" "L2a backacatter" → please specify which of the product(s) they used in the references (SCA, MIE, ICA, AEL-PRO or SCA-MLE)
L. 174 "L2A particle backscatter" → "SCA particle backscatter"
L. 273 "Aeolus L2A product " → change to "SCA product"
L. 322 "The L2A optical properties product which will be described in the next section, derived by the so-called Standard Correct Algorithm (SCA) (Flament et al., 2021), are provided at the observation scale (on a horizontal resolution of ~90 km) and are available through the Aeolus Online Dissemination System (https://aeolus-ds.eo.esa.int)." → The SCA is part of the L2a, but not the whole L2a is derived by the SCA algorithm. Please change to e.g. "The SCA optical properties are part of the L2A product which will be described in the next section, and are derived by the so-called Standard Correct Algorithm (SCA) (Flament et al., 2021). They are provided at the observation scale (on a horizontal resolution of ~90 km) and are available through the Aeolus Online Dissemination System (https://aeolus-ds.eo.esa.int)."
L. 359 "L2a extinction retrievals" → change to "SCA extinction retrievals"
L. 428 "Aeolus L2A products" → may be changed to "Aeolus SCA optical properties product"
L. 602 "L2a backscatter profiles" → "SCA backscatter profiles"
L. 616 "Aeolus L2A profiles" → "Aeolus SCA profiles"
L. 624, 641, 651, 652, 659, 880, 882, 923, 985, 1081, 1138  "L2A" → "SCA"

L. 39 "Aeolus profiles" → "SCA profiles" or "L2a profiles"
L. 45 "Aeolus performance" → "SCA performance"
L. 48 "Aeolus profiles" → "SCA profiles"
L. 53 "Aeolus performance" → "SCA performance" (though this is likely an issue with all the retrieval algorithms)
L. 170 "Aeolus backscatter profiles" → lease specify which of the product(s) they used in the references (SCA, MIE, ICA, AEL-PRO or SCA-MLE) instead of "Aeolus backscatter profiles"
L. 365, 626, 627, 658, 664, 816, 822, 877, 938, 943  "Aeolus" → "SCA"
L. 716, 730 "Aeolus" → "Aeolus SCA product"
L. 814, 1146 "Aeolus' performance" → "SCA performance"
L. 816, 826, 867,1053 "Aeolus" → "the SCA retrievals"
L. 842, 845 "Aeolus" → "the SCA "
L. 883 "Aeolus satellite (SAT) backscatter coefficient" → "Aeolus satellite (SAT) SCA backscatter coefficient"
L. 1002, 1024, 1025, 1027, 1039, 1042, 1050, 1163, 1164 "Aeolus" → "SCA"

---

## Author Response (AR2)

**Replies to the Reviewer #2**

We would like to thank the Reviewer for this second round of comments. We believe that the revised manuscript addresses adequately the two points raised by the Reviewer. Please see below our replies (regular font) to your comments (bold font).

**General reply**

**I would like to cordially thank the authors for their efforts and their detailed response. They made a great effort to restructure parts of the manuscript and to clarify the open questions. The earlier concerns on the cilmatological AOD analysis can be dropped based on the provided explanations and the additional data in the manuscript and supplement (coefficient of variation, spatial correlation and textual changes). Also, the textual changes regarding the 4 specific cases are appropriate, now being less conclusive and leaving open the possibility to noise-induced mismatches between SCA retrievals. I was particularly glad to see the negative values included in the Figure 3, which complement this discussion. However, two of the raised points have not been addressed to full satisfaction, but both of which will require mainly changes within the text. Hence, I recommend a minor review.**

**Point 1: Regarding the colocation criterion**

**Thanks for agreeing that it would be better to use the BRC center instead of the BRC start for colocation. I want to thank the authors for the effort of giving a detailed overview of the number of BRCs that are not affected: It is good to know that 77% of the BRCs (observations) are correctly colocated, which makes me confident that the reached conclusions are not too sensitive to any changes. However, below a discussion of open points:**

**"*On the contrary, in Athens, due to the "peculiarity" of the site such decision would exclude most of the matchups between Aeolus and ground-based profiles since ALADIN track resides near the edge of the defined circle.*"**

**I disagree with the statement that a change from the starting coordinate to the center coordinate will effectively reduce the number of BRCs.**

**To illustrate this point, a small graph below with some randomly simulated overpasses (Fig 1): Panel A shows the colocation method as is, illustrating the skewness around the circle in flight direction because of using the starting coordinate. Panel B resembles the procedure that the authors have shown in their rebuttal: Using the starting coordinate for colocation in the first place, but then checking subsequently whether also the center falls inside the circle. This method of course reduces the number of BRCs, because both the center and the starting coordinate are required to fall within the circle. In line with what the authors state, about 41/53 = 77% of the observations were correctly colocated in this example and remain. Nevertheless, panel C shows what happens using the center**

**coordinate right away, which results in a non-skewed choice of BRCs around the circles center.**

**On average, the version A and C will result in the same number of BRCs, because the density of observations is not reduced, just different BRCs are considered. However, in this random realisation, there are of course small fluctuations allowing for small differences (53 vs 52).**

*"Therefore, we think that it is better to proceed with our initial approach trying not to reduce further the already limited number of cases and BRCs."*

**Following my point above, this cannot be the motivation for keeping the flawed colocation method, as the number of BRCs is not reduced by changing to the center coordinates (since other BRCs that were previously disregarded are "entering the game" from below). Nevertheless, since the 77 % of the BRCs are correctly colocated, I do not expect a huge impact on the statistical analysis and the reached conclusions. So, although I am not agreeing with the authors, redoing the analysis with the correct colocation is probably a disproportional effort compared to the potential gain. I strongly recommend however, that since the authors conclude only satisfactory performance of Aeolus, they should stress clearly the weakness of their applied colocation criterion in the methods section. A reader might argue that any poor performance might be caused by the ~45 km offset, which allows in fact observations at 165 km distance instead of 120 km.**

[Figure]

Fig 1: Illustration of the differences between colocation methods on a random realisation of overpasses (the same realisation is used for panels A, B, C). The panels show the results of different colocation strategies. The satellite is passing from south to north in this example. The color-code follows the Figure 2 of the manuscript.

We have added the following paragraph in the revised manuscript.

*"Following this approach there is a possibility of including BRCs where more than half of their length to fall outside of the defined circle. This might affect the evaluation outcomes because we are not considering the BRC center in the collocation. Nevertheless, we are expecting a negligible impact on the statistical analysis since the 77% of the BRCs would have been selected using alternatively the coordinates at their center."*

**Point 2: Differentiating between "Aeolus", the "L2a product" and the "SCA product (or retrieval)"**

I earlier made the remarks:

*"Throughout most of the text, the authors do not differentiate between the performance of the Aeolus satellite itself and the performance of the retrieved SCA co-polar backscatter coefficient within the L2A product. This needs to be clarified, particularly since two significantly improved optical properties products are available as of March this year (see specific comments)."*

and

*">the most important finding is that Aeolus is not capable to reproduce satisfactorily the backscatter profiles< I find this a bolt statement to make here. It is not Aeolus but specifically the current Aeulos SCA product in absence of cloud flagging. "*

To which the authors answered:

*"As it is explained below, we have modified the relevant parts as suggested by the reviewer."*

Overall, this has not been adjusted throughout the manuscript so I explain below in a more detailed manner: What I meant to stress is that sentences like "The most important finding is that Aeolus is not capable to reproduce satisfactorily the backscatter profiles." (L.945 , manuscript with tracked changes) are not correct. Such statement suggests and concludes incapability of the Aeolus mission itself, while it is particularly the shortcomings of the SCA product that the authors want to underline. Hence, the above statement should be changed to either:

"The most important finding is that Aeolus' SCA product is not capable to reproduce satisfactorily the backscatter profiles."

 Or

"The most important finding is that the SCA retrieval is not capable to reproduce satisfactorily the backscatter profiles."

I stress this point particularly, since the science teams are and have been working on more refined L2a backscatter coefficient profiles that are already released and included into the L2a product. Hence, there is a need for specification. This is where my impression originated from, that the authors generalize a lot in sections 6.1.x, though I am now sure that this is certainly not intended and hence a misunderstanding due to choice of words. I strongly encourage the authors to differentiate, so all statements that state "L2a product" and "Aeolus" in place of "SCA product" need to be changed accordingly.

**To provide an extensive list:**

**L. 39 "L2A backscatter coefficient" → change to "SCA backscatter coefficient"**

Done

**L. 165, 950 "L2A optical properties" "L2a backacatter" → please specify which of the product(s) they used in the references (SCA, MIE, ICA, AEL-PRO or SCA-MLE)**

Done

**L. 174 "L2A particle backscatter" → "SCA particle backscatter"**

Done

**L. 273 "Aeolus L2A product " → change to "SCA product"**

Done

**L. 322 "The L2A optical properties product which will be described in the next section, derived by the so-called Standard Correct Algorithm (SCA) (Flament et al., 2021), are provided at the observation scale (on a horizontal resolution of ~90 km) and are available through the Aeolus Online Dissemination System (https://aeolus-ds.eo.esa.int)." → The SCA is part of the L2a, but not the whole L2a is derived by the SCA algorithm. Please change to e.g. "The SCA optical properties are part of the L2A product which will be described in the next section, and are derived by the so-called Standard Correct Algorithm (SCA) (Flament et al., 2021). They are provided at the observation scale (on a horizontal resolution of ~90 km) and are available through the Aeolus Online Dissemination System (https://aeolus-ds.eo.esa.int)."**

Done

**L. 359 "L2a extinction retrievals" → change to "SCA extinction retrievals"**

Done

**L. 428 "Aeolus L2A products" → may be changed to "Aeolus SCA optical properties product"**

Done

**L. 602 "L2a backscatter profiles" → "SCA backscatter profiles"**

Done

**L. 616 "Aeolus L2A profiles" → "Aeolus SCA profiles"**

Done

**L. 624, 641, 651, 652, 659, 880, 882, 923, 985, 1081, 1138 "L2A" → "SCA"**

Done

**L. 39 "Aeolus profiles" → "SCA profiles" or "L2a profiles"**

Done

**L. 45 "Aeolus performance" → "SCA performance"**

Done

**L. 48 "Aeolus profiles" → "SCA profiles"**

Done

**L. 53 "Aeolus performance" → "SCA performance" (though this is likely an issue with all the retrieval algorithms)**

Done

**L. 170 "Aeolus backscatter profiles" → lease specify which of the product(s) they used in the references (SCA, MIE, ICA, AEL-PRO or SCA-MLE) instead of "Aeolus backscatter profiles"**

Done

**L. 365, 626, 627, 658, 664, 816, 822, 877, 938, 943 "Aeolus" → "SCA"**

Done

**L. 716, 730 "Aeolus" → "Aeolus SCA product"**

Done

**L. 814, 1146 "Aeolus' performance" → "SCA performance"**

Done

**L. 816, 826, 867,1053 "Aeolus" → "the SCA retrievals"**

Done

**L. 842, 845 "Aeolus" → "the SCA "**

Done

**L. 883 "Aeolus satellite (SAT) backscatter coefficient" → "Aeolus satellite (SAT) SCA backscatter coefficient"**

Done

**L. 1002, 1024, 1025, 1027, 1039, 1042, 1050, 1163, 1164 "Aeolus" → "SCA"**

Done

**Replies to the Reviewer #4**

We would like to thank the Reviewer for his/her comments helping us to clarify better critical points of our analysis. Please see below our point-by-point replies (regular font) to the comments (bold font) raised by the Reviewer.

**The authors responded to most of the reviewers' comments. There was an effort to make the paper lighter but long, heavy sentences, of the kind you need to read twice to be sure you understand, are still present. I propose some simplifications in the specific comments.**

We have made an effort to simplify further "heavy" sentences in the manuscript.

**Following reviewer #2, I think it is important to mention that the SCA mid-bin backscatter is not expected to perform better than the normal SCA, the mid-bin algorithm is designed to reduce the extinction errors. The authors are aware of this, because it is mentioned on l. 397-405. But the comparison of performance between the two results is still at the heart of the paper. I am not sure the paper is clear enough on this point. From his/her comments, it seems that even reviewer #1 misunderstood this.**

In the revised manuscript we are clarifying better this point.

**Also, it is important to note that the SCA performance is not the performance of Aeolus itself. It's one of the optical properties products and, hopefully, there is still room for improvement.**

We have corrected all the relevant instances in the revised manuscript.

**Getting back to the main subject of the paper, the validation of satellite observations from ground-based measurements, I think the work is useful and presents an important contribution to a difficult problem. A meta-analysis, exploiting more numerous profiles from more locations would be very useful (e.g. co-ordinated at EARLINET level? Ok, it is mentioned in the conclusion!).**

We are already mentioning the ongoing work of the Aeolus EARLINET Cal/Val study.

**On the scientific side, I think most limitations of the study are properly acknowledged.**

Thanks!

**Specific comments:**

**l. 27: "along with wind HLOS profiles" could be removed to make the sentence lighter. Or simply replaced by "also".**

Done.

**l. 31 "(capital of Greece)" the administrative status is not a geographical indication. Centre of Greece?**

We have replaced "capital of Greece" with "central Greece".

**Several occurrences in the text of "the performance downgrades" (l.45, l.53, l.1157). I would replace "downgrade" with "degrade". I am being picky but this bothered me. You wouldn't use "upgrade" if the performance improved, would you?**

We have replaced all the relative instances in the manuscript as suggested by the reviewer.

**l. 287 named by -> named after?**

Done

**l.295 The orbital velocity is actually closer to 7.7 km/s**

Thanks for the correction!

**l.298: The precision that solar panels are facing the sun is not useful. If you want to keep this sentence, you should focus on the telescope rather than on the panels. Something like "The telescope is pointed to the right of the flight direction, aiming into the night hemisphere". As you are not discussing the influence of solar background signal, you could omit this sentence.**

We have modified this sentence as follows: "*Aeolus is flying over the terminator between day and night (dawn/dusk orbits), with its telescope pointing to the right of the flight direction (aiming into the night hemisphere) for minimizing the solar background illumination (Kanitz et al., 2019).*"

**l. 300 "HSRL lidar" contains lidar twice. "HSRL" alone or "HSR lidar"?**

Done

**l. 309 As you want to focus on the description of the geometry here, you may simply say that "Aeolus provides observations at the slant nadir angle of 35 degrees".**

We prefer to keep our description.

**l.315 You could use a simpler structure: "Therefore, the discrimination between aerosol and clouds, as well as the typing of aerosols, is challenging".**

We have modified this part of the text as follows: "*However, ALADIN only measures the co-polar part of the atmospheric backscatter and at a single wavelength. Therefore, the discrimination between aerosols and clouds and their respective subtypes is challenging.*"

**l. 336: I would call the product "(particulate) optical properties product", to stress that it focusses purely on optical properties, without any classification or any chemistry consideration.**

We believe that it is pretty clear and there is not any misunderstanding here.

**l. 344: Is it correct to replace "backscatter (either molecular or particulate) where the squared one-way transmission through the atmosphere is taken into account" by "attenuated backscatter"?**

Not yet because it has not been introduced in the text the cross-talk correction, the normalization by the range bin thickness and the correction by the squared range.

**l. 346-354: I am not sure that this description of the cross-talk correction with the coefficients name is needed. This paragraph could probably be summarized in one sentence.**

We believe that it is better to keep this short paragraph.

**l. 354-405: This is a long introduction to explain the interest of the "SCA mid-bin" product. As a reader, I would prefer to be pointed to the ATBD or to Flament et al. 2021. It would save one page of reading and allow the reader to focus on the actual comparison.**

We believe that it is better to provide this information (shortly mentioned) in the text.

**l. 411:**

**- Do you have a reference on comparisons of the MLE to ground based lidars? Ehlers et al. only provide comparisons with simulated and CALIPSO data.**

There are some first comparisons versus eVe at Cabo Verde shown in Aeolus meetings.

**- Also, the sentence spanning l. 410 to l. 413 is a long one again. "Comparison against ground based observation showed that the precision of the extinction and LR retrieved by the MLE is much better than the one from the SCA".**

We think that it is better to keep this sentence as is.

**- What are "SCA end-to-end simulated optical products"? It is not clearly described.**

We are copying from Ehlers et al. (2022) and we are providing for further details the work of Reitebuch et al. (2018)

*"The simulated data are produced with the Aeolus end-to-end simulator described in Reitebuch et al. (2018b), which allows realistic simulations of ALADIN measurements from defined atmospheric scenes as input to the L1B algorithm. Its output data are provided in the same format and temporal and spatial resolution as nominally downlinked from the satellite in order to test the whole processing chain up to the optical properties delivered in the L2A product. The simulation covers the charge transfer and detection on the accumulation charge-coupled device (ACCD) including offsets, non-linearity and noise sources, such as dark current noise, read-out noise, Poisson detection noise (shot noise) and the analogue-to-digital conversion with 16 bit."*

Reitebuch, O., Marksteiner, U., Rompel, M., Meringer, M., Schmidt, K., Huber, D., Nikolaus, I., Dabas, A., Marshall, J., de Bruin, F., Kanitz, T., and Straume, A.-G.: Aeolus End-To-End Simulator and Wind Retrieval Algorithms up to Level 1B, EPJ Web Conf., 176, 02010, https://doi.org/10.1051/epjconf/201817602010, 2018.

**l. 566: It is not clear to me whether this is the standard deviations with respect to time or space. I would think time.**

In the previous sentence we are discussing about the temporal variability. Nevertheless, a better explanation is given in the revised text.

**l. 588 "cycles" should be "circles"?**

Corrected. Thanks!

**l.597-598: "strong horizontal variability" fig 1-iii, 1-iv and S1 say the opposite: strong spatial homogeneity around Antikythera (From S1-ii, I understand that the time series of average AOD within each circle are very strongly correlated in ANT. So, the average within any given circle is close to the average within another one, most of the time.)**

We have modified this sentence in the revised manuscript.

**l.611 and Figure 2: the green arrow is not much better than the orange one: we cannot see in which direction it is pointing (Maybe because of image compression in the manuscript I received?). It doesn't really matter, all the tracks pictured are descending anyway, this can be said in the text.**

Done.

**l. 663 "entire" should be replaced by "entirely"**

Done.

**l.721 "it is revealed" doesn't sound idiomatic**

We have replaced "revealed" with "evident".

**l. 722-747 I wouldn't use "underestimated" (l.723) or "negative biases" (l.730) which suggests a deficiency of the Aeolus instrument or processing. It is only a deficiency in the nomenclature, I agree that the variables in the L2A should probably be named "circular co-polar backscatter" but we should focus on comparing apples with apples.**

**Here, it is just a matter of presentation: I would explain the importance of the correction first, and then present the results.**

We have slightly modified this paragraph.

**l.816-819: "due to overlying noise (i.e. negative backscatter)": I don't agree with this explanation. If you mean that errors from the overlying layers are propagated downwards, this is wrong. Within any profile, backscatter retrievals of the SCA are independent from each other.**

We have replaced this sentence and the next one with:

*"SCA fails to reproduce the aerosol layer (in terms of structure and backscatter magnitude) seen from the ground-based lidar between 2 and 4 km."*

**l.841-842: I do not see the "layer" between 5.5 and 8 km on Figure 3-iv. I think it is important to stress that we should not over-interpret single profiles. The noise is very large, and errors are underestimated.**

We have modified the sentence as follows:

*"As a result, Aeolus possibly (acknowledging the weak signals and the underestimated errors) detects incorrectly an aerosol layer between 5.5 and 8 km under the assumption that clear-sky conditions are appropriately represented in the MSG-SEVIRI imagery and remain constant within the time interval (~6 minutes) of MSG and Aeolus observations."*

**l. 947: This mostly shows that thin range bins close to the ground are detrimental to optical properties retrieval because of the low SNR they induce (cause (ii) below). The signal from the surface should only leak into the first bin above the ground.**

We are providing sufficient interpretation throughout the text.

**l. 964-981: How many profiles are used for comparison after filtering?**

Initially we are analyzing 82 profiles and after cloud filtering 36.

**Figure 5 and this paragraph really underline the importance of proper cloud filtering. The results of figure 4 are only useful to show the contrast.**

This is exactly our intention and for this reason we are contrasting the vertically resolved evaluation metrics with and without the consideration of cloud-contaminated profiles.

**l. 1078-1080: Out of curiosity: how invalid can the underlying assumptions be? Is there a quantitative assessment somewhere? That would also be an important step for later studies, because few ground lidars actually have the circular polarization capability (could this assessment be a task for eVe?).**

This task can be addressed by eVe which is capable of measuring both linear and circular backscatter. The first eVe measurements acquired in Cabo Verde in the framework of the ASKOS campaign show that the linear-circular deviations are small or negligible. Some preliminary results are shown here (it is required a registration to the Aeolus confluence page).

**l. 1100-1102: this consideration was not discussed before and only appears here.**

This is not a major issue in our study but it is an aspect which should be considered in Cal/Val studies at stations located at areas with highly variable topography.

**L. 1141: again this "underestimation" is more a problem of misinterpretation.**

We have modified the sentence as follows: *"The misdetection of the cross polarized lidar return signals can interpret the lower Aeolus SCA backscatter values (ranging from 13% to 33%) with respect to ground-based retrievals when depolarizing mineral particles are probed (case of 10th July 2019)."*

**l. 1159: why not say: "Our analysis reveals that …"?**

Please see below the modified sentence.

*"Our statistical assessment analysis reveals that the removal of cloud contaminated spaceborne profiles, achieved via the synergy with MSG-SEVIRI cloud observations, results in a significant improvement of the product performance."*

**Figure 1: the legend doesn't contain a description for sub-figure 1-v**

We have added a description for Figure 1-v. Thank you for the correction!